# Salt-stress-induced tomato sweetening involves an SlSnRK2.6-SlZHD8 sugar accumulation cascade triggered by root-derived abscisic acid

Jinghao Xu[1,6], Zhiliang Zhang[2,6], Jin-Wei Wei[3,6], Yingfang Zhu [ID][4,6], Dan Zhao[1], Tianchen Xia [ID][1], Xiaoqian Liu[1], Chengqiang Wang[5] & Biao Gong [ID][1✉]

## Abstract

**Crop quality arises from the interplay of genetics and environment. While moderate salt stress is known to enhance fruit sweetness, the underlying molecular mechanisms remain unclear. Using tomato (*Solanum lycopersicum*) as a model, this study investigates how salt stress promotes fruit sugar accumulation. Root-derived abscisic acid (ABA) transport to fruit acts as the key signal under salt stress. Elevated fruit-ABA activates the kinase SlSnRK2.6, which phosphorylates the SlZHD8 transcription factor. This phosphorylation inhibits SlZHD8 function by reducing its protein stability and DNA-binding, thereby relieving its repression of SlSUS3 and SlSWEET12 to enhance fruit-sugar accumulation. Furthermore, the SlSnRK2.6-SlZHD8-SlSWEET12 module also regulates root-sugar accumulation and confers salt tolerance. Evolutionary analysis revealed a beneficial ZHD8 haplotype, whose reduced promoter-binding affinity promotes fruit-sugar accumulation under normal conditions and enhances salt tolerance. These findings explain how stress enhances quality and highlight the potential of key mutations of ZHD8, particularly the beneficial haplotype, for breeding tomatoes with improved sugar content and salt tolerance.**

**Keywords** ABA Signaling; Domestication; Salt Tolerance; Sugar Metabolism; Tomato
**Subject Category** Plant Biology

## Introduction

Saline and saline-sodic soils cover 831–932 million hectares globally. High soil salinity adversely impacts both crop growth and yield. However, an old Chinese adage states: "顺境出产量, 逆境促品质", or "favorable conditions bring yield, stress enhances quality". This implies that while favorable conditions maximize yields, moderate environmental stress, particularly salt and drought stress, can improve crop quality. In horticultural crops, the enhancement of quality traits under stress is often observed, being particularly prevalent in high-sugar-content species. Despite inhibiting growth and yield, moderate salt stress is deliberately applied through practices like micro-saline water irrigation, elevated electrical conductivity (EC) in nutrient solutions, or cultivation in naturally slightly saline-alkali soils. These tillage methods enhance quality and market value in crops like tomatoes (De Pascale et al, 2001), melons (Botía et al, 2005), and grapes (Permanhani et al, 2016). Additionally, in regions facing freshwater scarcity, micro-saline water irrigation can partially meet crop water demands and maintain reasonable yield levels (Li et al, 2019). Nevertheless, the molecular mechanisms by which salt stress enhances fruit sweetness remain unclear. Addressing these questions is crucial for theoretically decoupling the negative correlation between growth inhibition and quality formation under salt stress conditions.

Tomato (*Solanum lycopersicum*) serves as a model plant for studying fruit development and quality traits, particularly sugar content. It exhibits good salt tolerance (Cuartero et al, 2006; Gong et al, 2013) and has a successful cultivation history and industrial foundation in mildly saline soils (salt content <0.3%). Tomato fruit sugars are predominantly glucose and fructose, constituting 55–65% of the total soluble solids, while significant sucrose accumulation is limited to only a few varieties (Colantonio et al, 2022; Zhang et al, 2024). The phloem transport of leaf-derived sucrose to fruits is mediated by two key transporter families: sucrose transporters and sugars will eventually be exported transporters (SWEETs). Within fruits, sucrose is hydrolyzed into glucose and fructose by sucrose synthases (SUSs) and invertase (Quinet et al, 2019). Sugar content of tomatoes is highly susceptible to the salt concentration of the cultivation medium. In the 1990s, Japan pioneered high-sugar tomato production using salt-regulation technology, initially attributed to a simple "concentration effect." Subsequent research revealed that salt stress also modulates complex genetic and metabolic networks in tomato fruits. This modulation not only increases fruit sugar content but

[1]College of Horticulture Science and Engineering, Shandong Agricultural University, 271018 Taian, China. [2]Institute of Crop Sciences, Chinese Academy of Agricultural Sciences, 100101 Beijing, China. [3]Institute of Genetics and Developmental Biology, Chinese Academy of Sciences, 100101 Beijing, China. [4]School of Life Sciences, Henan University, 475001 Kaifeng, China. [5]College of Life Sciences, Shandong Agricultural University, 271018 Taian, China. [6]These authors contributed equally: Jinghao Xu, Zhiliang Zhang, Jin-Wei Wei, Yingfang Zhu. ✉E-mail: gongbiao@sdau.edu.cn

also promotes the accumulation of compounds like malic acid and sweet amino acids, which enhance human taste sensitivity to sweetness (Tang et al, 2020; Liu et al, 2024a). These findings demonstrate that salt stress enhances sugar accumulation in tomato, going beyond passive concentration effects to involve active physiological and regulatory adjustments. The underlying biological significance and potential signaling pathways warrant careful investigation.

For humans, sugar signifies nutrition and quality. For plants, it represents energy and survival. Under salinity or drought, plants reshape sugar partitioning between sources and sinks. This shift ultimately supports survival and growth through coordinated osmotic and adaptive responses (Thalmann et al, 2016; Rodrigues et al, 2019; Chen et al, 2022). Abscisic acid (ABA) may be integral to this regulatory network, playing a pivotal role in both plant salt tolerance and in the induction of tomato fruit expansion and ripening (Zou et al, 2022). ABA accumulation in tomato fruits initiates during the expansion stage, peaks at the breaker stage, and subsequently declines (Sun et al, 2012). Its function includes establishing strong fruit sink strength and inhibiting seed dormancy (Shohat et al, 2022). Based on existing literature, the accumulation curves of ABA and sugar during tomato fruit ripening are largely parallel (Kanayama, 2017; Sun et al, 2012; Zhang et al, 2024). This observation leads to the hypothesis that ABA may be a key signaling molecule mediating fruit sugar accumulation under salt stress. Under salt stress, $Na^+$ primarily accumulates in tomato roots, stems, and leaves, while its concentration in fruits and seeds remains minimal (Zhang and Blumwald, 2001). This indicates that fruit cells lack direct perception of salt stress. In rice, leaves have been shown to act as temperature sensors that synthesize and transport ABA to the seeds to regulate their development (Qin et al, 2021). Current opinion supports that plants employ systemic ABA signaling to coordinate responses to adverse environmental conditions (Daszkowska-Golec, 2022). Ultimately, the molecular mechanism of ABA involvement in stimulating fruit sugar accumulation under salt stress, as well as the site of ABA synthesis, remains unclear.

The rate-limiting enzyme for ABA synthesis is 9-cis-epoxycarotenoid dioxygenase (NCED). It is primarily located in the vascular tissues of roots, leaves, fruits, and seeds (Boursiac et al, 2013). This localization facilitates ABA's long-distance transport and its role in regulating fruit and seed development (Daszkowska-Golec, 2022). Upon reaching target tissues, ABA is perceived by receptors, designated as regulatory components of ABA receptor (RCAR), pyrabaction resistance 1 (PYR1), or PYR1-like (PYL), and clade a protein phosphatases of type 2C (PP2C), which function as coreceptors. The binding of ABA to its receptors/coreceptors alleviates the inhibition of PP2C on sucrose non-fermenting-1-related protein kinase 2s (SnRK2s). Activated SnRK2s then directly phosphorylate downstream proteins or activate ABA-responsive element (ABRE)-binding proteins (AREBs)/ABRE-binding factors (ABFs) transcription factors or other targets to induce downstream gene expression, completing ABA signal transduction (Yoshida et al, 2019). Two direct pieces of evidence identify SnRK2s as potential mediators of ABA-regulated sugar metabolism during both fruit ripening and stress responses. In apple fruit, SnRK2.3 phosphorylates AREB1 to transcriptionally regulate *tonoplast sugar transporter 1/2*, promoting sugar accumulation (Zhu et al, 2023). Additionally, phosphorylation of SWEET11/12 by SnRK2.2/2.3/2.6

enhances the root/shoot ratio in drought-stressed *Arabidopsis* to acquire stronger water adaptation capabilities (Chen et al, 2022). Additionally, SnRK2.6 phosphorylates the flowering-related VOZ1 transcription factor vascular, thereby promoting early flowering in drought-stressed tomato (Chong et al, 2022). Consequently, we propose that the ABA signaling pathway plays a key molecular role in regulating energy metabolism to coordinate growth-stress tradeoffs. However, whether these processes contribute to the "stress enhances quality" phenomenon and their precise functional mechanisms require further investigation.

Here, we investigated tomato responses under moderate salt stress, where salt stress enhances fruit quality without seriously compromising yield. We identified root-derived ABA accumulation and its subsequent transport to fruits as a key physiological process promoting fruit sugar accumulation. Within fruits, ABA activates SlSnRK2.6, which phosphorylates the zinc-finger homeodomain protein 8 (SlZHD8) transcription factor. SlZHD8 acts as a transcriptional repressor of the sugar accumulation genes *SlSUS3* and *SlSWEET12*. Phosphorylation by SlSnRK2.6 impairs SlZHD8's protein stability and DNA-binding capacity, thus relieves transcriptional repression of *SlSUS3* and *SLSWEET12*, thereby promoting fruit sugar accumulation. Furthermore, SlSWEET12 functions as a critical sucrose transporter from shoots to roots. This enhanced sugar supply supports root growth and energy metabolism, ultimately improving tomato salt tolerance. Collectively, the *SlSnRK2.6-SlZHD8-SlSUS3/SlSWEET12* module represents an effective molecular mechanism that coordinately enhances fruit sweetness and salt tolerance. Additionally, we identified different haplotypes in *ZHD8* within tomato domestication populations, revealing its potential application value for molecular breeding. These findings provide novel insights into crop environmental adaptation and quality regulation, offering promising targets for breeding improvement.

## Results

### Salt stress triggers fruit sugar accumulation along with ABA elevation

To guarantee economic returns, sweet tomatoes are commonly cultivated under moderate salt stress. We initially investigated the effects of 0.1–0.3% NaCl on tomato plant growth, fruit yield, and sugar contents (represented by contents of soluble solids, glucose, and fructose; the same below). Salt stress exerted a concentration-dependent inhibitory effect on plant growth and fruit yield (Appendix Fig. S1A,B). However, all three stress concentrations significantly increased fruit sugar contents, with no significant differences between them (Appendix Fig. S1C–E). Consequently, we selected the 0.1% NaCl treatment, where growth and yield losses both remained below 10%, as the moderate salt stress for subsequent experiments.

Specifically, salt stress reduced fruit yield (by ~8.41%) and accelerated fruit ripening (Fig. 1A–C). We measured sugar and ABA contents in fruits at different ripening stages, defined according to standard classification criteria (Li et al, 2020). As ripening progressed, sugar contents followed a unimodal curve, peaking at turning stage (Fig. 1D–F). Salt stress increased sugar contents across ripening stages, with the most significant difference

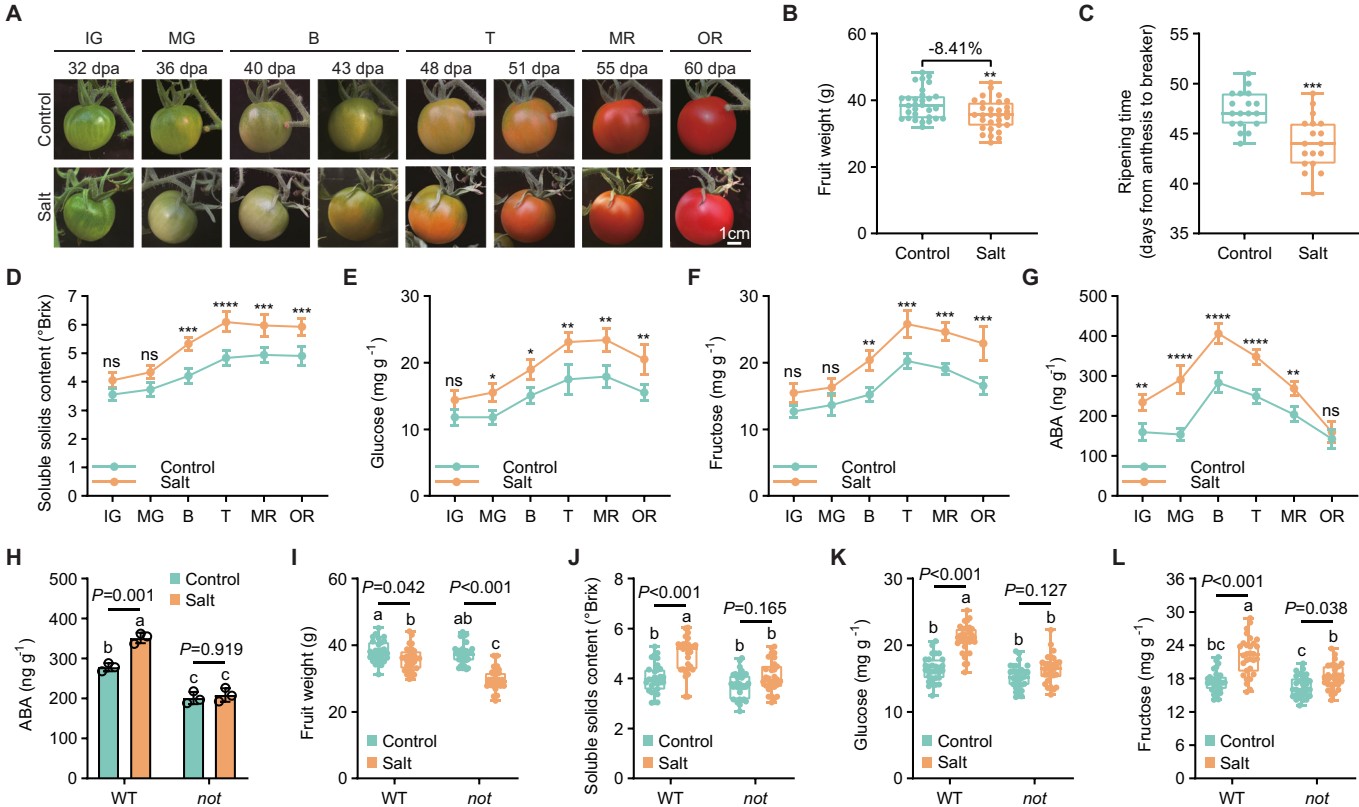

**Figure 1. Elevated ABA mediates fruit sugar accumulation during salt stress.**

(A–G) Influence of salt stress on (A) fruit phenotype, (B) fruit weight, (C) ripening time, (D) soluble solids content, (E) glucose content, (F) fructose content, and (G) ABA content. Fruit samples of WT plants under control and salt stress were used for test. Ripening process classification: the fruit phenotype from 32 to 60 days post anthesis (dpa) was divided into immature green (IG), mature green (MG), breaker (B), turning (T), mature red (MR), and over red (OR). Box plots of (B) ($n = 30$) and (C) ($n = 18$) show median with 0.25 and 0.75 quartiles, whiskers represent values from minimum to maximum; data of (D–G) represent mean ± SD ($n = 3$); statistical significance was determined by $t$ test or two-way ANOVA ($^{ns}P > 0.05$, $*P < 0.05$, $**P < 0.01$, $***P < 0.001$, $****P < 0.0001$). (H–L) The interplay between endogenous ABA and salt stress affecting (H) ABA content, (I) fruit weight, (J) soluble solids content, (K) glucose content, and (L) fructose content. Turning-stage fruits of WT and *not* in control and salt stress were used for test. Data of (H) represent mean ± SD ($n = 3$); box plots of I to L show median with 0.25 and 0.75 quartiles, whiskers represent values from minimum to maximum ($n = 30$); statistical significance was determined by one-way ANOVA (different letters indicate significant differences at $P < 0.05$). Source data are available online for this figure.

observed at turning stage (Fig. 1D–F). While ABA levels exhibited a similar unimodal pattern to sugar, its peak occurred earlier, at breaker stage (Fig. 1G). This implies that ABA may play a role in mediating stress-induced sugar accumulation. Given that salt stress alters ripening timing, it is more appropriate to compare fruits at equivalent ripening stages rather than at the same number of days post anthesis (dpa). Due to the maximal difference in sugar contents between salt and control treatments occurring at turning stage, fruit samples at this stage were selected for subsequent studies.

## Elevated ABA mediates fruit sugar accumulation during salt stress

To investigate whether ABA mediates fruit sugar accumulation, we treated fruits with 1–15 mM ABA (25 μL per fruit) at the immature green stage and measured sugar contents at the turning stage. Increasing exogenous ABA concentrations significantly elevated fruit sugar accumulation (Appendix Fig. S2), demonstrating the direct role of ABA in promoting sugar accumulation. We

subsequently compared endogenous ABA level, fruit weight, and sugar contents between wild-type (WT) and the ABA synthesis mutant (*SlNCED1* mutation; *notabilis*, *not*) under both control and salt stress conditions. Compared to WT, the *not* mutant prevented salt-induced fruit ABA accumulation and exhibited reduced fruit weight (Fig. 1H,I). Crucially, while salt stress significantly increased sugar contents in WT fruits, this promoting effect was largely abolished in the *not* fruits (Fig. 1J–L). So, salt-induced ABA is crucial for promoting fruit sugar accumulation.

## Root-derived ABA mediates fruit sugar accumulation during salt stress

Root, leaf, and fruit tissues are all capable of synthesizing ABA. Salt stress specifically induced the expression of ABA biosynthesis genes (*SlNCED1* and *SlNCED2*) in roots and leaves, whereas transcript levels in fruit remained unchanged (Appendix Fig. S3A,B). However, ABA content increased significantly in root, leaf, and fruit during salt stress (Appendix Fig. S3C). This implies that fruit ABA accumulation likely results from translocation from leaf or

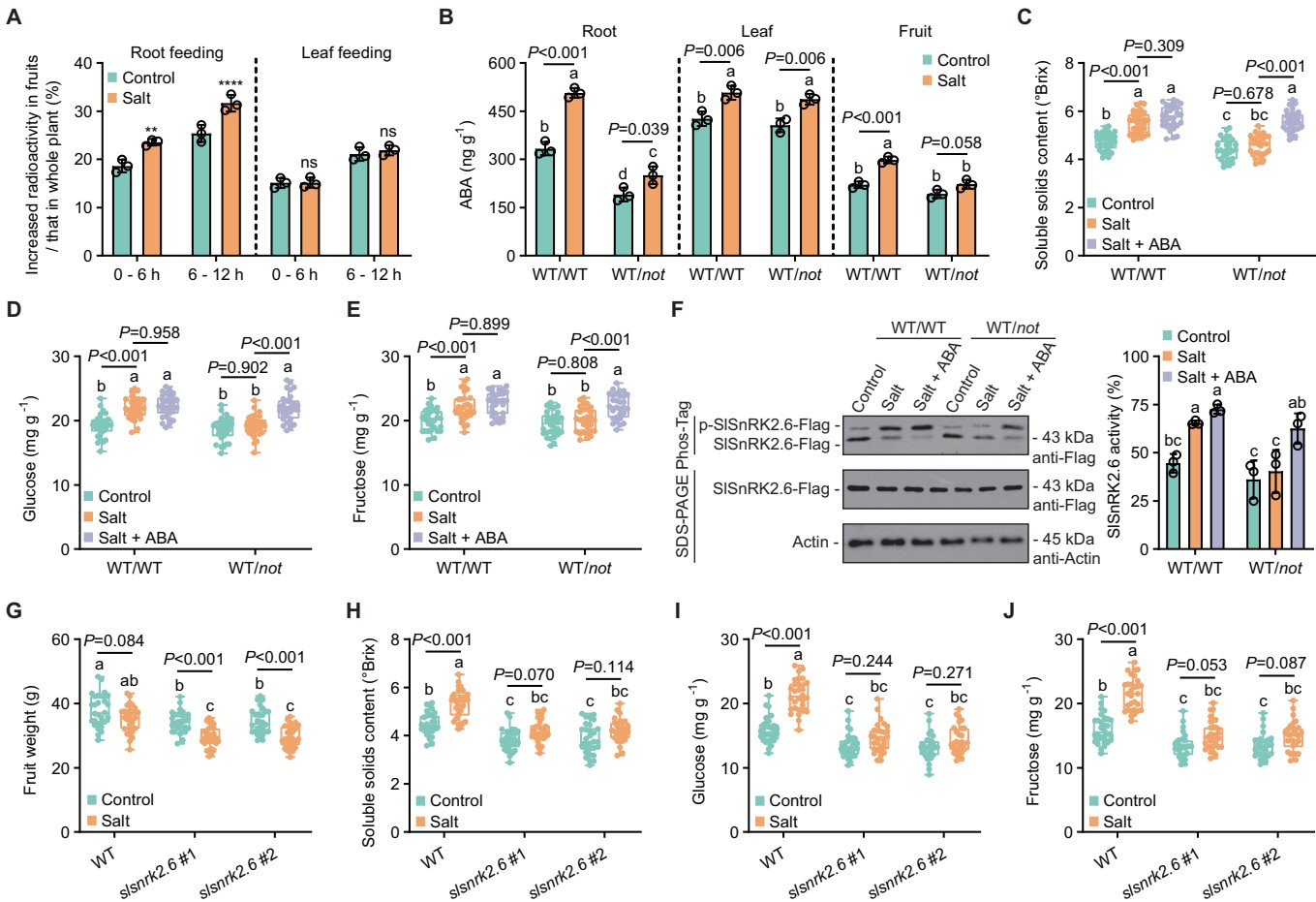

**Figure 2. Root-derived ABA triggers SlSnRK2.6 kinase to mediate fruit sugar accumulation during salt stress.**

(A) Influence of salt stress on ABA transport from root/leaf to fruit. The increased radioactivity in fruits were calculated for the 0-6 h and 6-12 h periods. Data represent mean ± SD ($n = 3$); statistical significance was determined by two-way ANOVA ($^{ns}P > 0.05$, $**P < 0.01$, $****P < 0.0001$). (B) ABA content of WT-grafted (WT/WT) and *not*-grafted (WT/*not*) plants in control and salt stress. Data represent mean ± SD ($n = 3$); statistical significance was determined by one-way ANOVA (Different letters indicate significant differences at $P < 0.05$). (C–F) The interplay between root-derived ABA and salt stress affecting (C) soluble solids content, (D) glucose content, (E) fructose content, and (F) SlSnRK2.6 kinase activity. Measurements used fruits from WT/WT and WT/*not* plants in control, salt, and salt + ABA treatments. For exogenous ABA treatment, fruits at immature green stage were uniformly injected with 25 μL aqueous solution of 10 mM ABA with the micro-syringe. Measurements were performed when the fruits reached the turning stage. Kinase activity (F) was assessed in PVX-mediated *SlSnRK2.6*-Flag transgenic fruits. Protein abundance and mobility shift were detected by immunoblot and Phos-Tag assay, respectively. The ratio of p-SlSnRK2.6-Flag is shown on the right of the Phos-Tag image. Box plots of (C–E) show median with 0.25 and 0.75 quartiles, whiskers represent values from minimum to maximum ($n = 30$); data of (F) represent mean ± SD ($n = 3$); statistical significance was determined by one-way ANOVA (Different letters indicate significant differences at $P < 0.05$). (G–J) The interplay between *SlSnRK2.6* and salt stress affecting (G) fruit weight, (H) soluble solids content, (I) glucose content, and (J) fructose content. Turning-stage fruits of WT and *slsnrk2.6* in control and salt treatments were used for test. Box plots show median with 0.25 and 0.75 quartiles, whiskers represent values from minimum to maximum ($n = 30$); statistical significance was determined by one-way ANOVA (different letters indicate significant differences at $P < 0.05$). Source data are available online for this figure.

root rather than de novo synthesis. To directly confirm ABA translocation from root or leaf to fruit and assess its promotion during salt stress, we performed isotope labeling experiments. We supplied ³H-ABA to roots or leaves of 'Micro-Tom' (a dwarf tomato variety ideal for experimental systems) and detected radioactivity in fruits. Radioactivity was observed in fruit under both feeding treatments, but salt stress enhanced the signal only when ³H-ABA was supplied to root (Fig. 2A). This demonstrates that salt stress specifically facilitates ABA transport from root to fruit.

To establish root-derived ABA as the key regulator of fruit sugar accumulation during salt stress, we grafted WT scions onto WT or *not* rootstocks, generating WT/WT and WT/*not* plants, respectively. Compared to WT/WT, WT/*not* exhibited root-specific ABA biosynthesis deficiency, especially under salt stress (Fig. 2B). Meanwhile, leaf ABA content showed no significant differences between graft types under both conditions (Fig. 2B). This grafting approach was employed to isolate the contribution of root-derived ABA to fruit sugar accumulation. Although fruit ABA content increased under salt stress in both graft combinations, the increase was markedly reduced in WT/*not* versus WT/WT, demonstrating root-derived ABA drives fruit ABA accumulation (Fig. 2B). Notably, salt-induced fruit sugar accumulation was markedly attenuated in WT/*not* relative to WT/WT (Fig. 2C–E). Meanwhile,

exogenous application of ABA to the fruits of WT/*not* plants restored the phenotype of defective response to salt stress (Fig. 2C–E). These results demonstrate root-derived ABA mediates fruit sugar accumulation during salt stress.

## SlSnRK2.6 mediates fruit sugar accumulation during salt stress

Since fruit ABA accumulation promotes sugar accumulation and the SnRK2 family functions as both the core ABA signaling component and a key energy regulator (Belda-Palazón et al, 2020), we investigated SlSnRK2s. A BLAST search of the tomato genome using *Arabidopsis* AtSnRK2s identified seven members (Appendix Fig. S4A). Analysis of *SlSnRK2s* expression in fruits showed distinct, salt stress-induced patterns throughout ripening (Appendix Fig. S4B). Among these, *SlSnRK2.3* and *SlSnRK2.6* exhibited the strongest induction, and their expression levels were significantly positively correlated with sugar content during maturation (Appendix Fig. S4B,C). To investigate their genetic function, we employed virus-induced gene silencing (VIGS) to suppress *SlSnRK2.3* and *SlSnRK2.6* in fruits. Silencing of *SlSnRK2.6* (TRV-*SlSnRK2.6*), but not *SlSnRK2.3* (TRV-*SlSnRK2.3*), significantly reduced salt stress-induced sugar accumulation compared with the empty vector (TRV-*EV*) control (Appendix Fig. S4D–H). We used Potato virus X (PVX) vectors to express SlSnRK2.6-Flag in fruits and employed immunoblot analysis alongside Phos-Tag assay to detect the kinase activity. During ripening, SlSnRK2.6 kinase activity markedly increased at the breaker and turning stages (Appendix Fig. S4I). This trend aligns with changes of ABA during fruit ripening (Fig. 1G), suggesting the activation of SlSnRK2.6 kinase by ABA accumulation. We further measured SlSnRK2.6 kinase activity in turning-stage fruits within the previously established grafting system. Salt stress significantly induced SlSnRK2.6 kinase activity in WT/WT fruits (Fig. 2F). In contrast, the activity in WT/*not* fruits was markedly lower but was restored to responsive levels by exogenous ABA treatment (Fig. 2F). We further generated two *slsnrk2.6* mutants using CRISPR-Cas9 technology. These mutants harbored 2-bp and 5-bp deletions, respectively, both resulting in premature termination of the encoded protein (Appendix Fig. S4J). Knocking out *SlSnRK2.6* significantly reduced fruit yield, with a more pronounced effect under salt stress (Fig. 2G). Crucially, compared to the WT, *slsnrk2.6* mutants almost completely abolished stress-induced fruit sugar accumulation (Fig. 2H–J). Collectively, root-derived ABA activates SlSnRK2.6 kinase to promote fruit sugar accumulation during salt stress.

## Screening identifies SlZHD8 as a SlSnRK2.6-interacting protein regulating sugar accumulation during salt stress

To investigate how SlSnRK2.6 regulates sugar accumulation, we performed a yeast two-hybrid (Y2H) screen using a cDNA library constructed from tomato fruits during salt stress. This screen identified 86 potential SlSnRK2.6-interacting proteins. However, gene function annotation revealed no candidates directly associated with sugar metabolism or transport among these candidates. Consequently, we focused on the five transcription factors identified in the screen (Appendix Fig. S5A), hypothesizing that they might be regulated by SlSnRK2.6 and subsequently influence

fruit sugar accumulation through transcriptional reprogramming. Using VIGS approach, we silenced each of these five transcription factors and assessed their effects on fruit sugar accumulation under both control and salt stress conditions. Notably, silencing the *SlZHD8* abolished stress-induced sugar accumulation (Appendix Fig. S5B–E). Under control condition, *SlZHD8*-silenced (TRV-*SlZHD8*) fruits accumulated more sugar compared to the TRV-*EV* fruits (Appendix Fig. S5C–E).

SlSnRK2.6 localizes to both the cytoplasm and nucleus (Chong et al, 2022), while SlZHD8 is nuclear-localized (Fig. 3A). This spatial overlap supports the possibility of in vivo interaction. Accordingly, we confirmed their in vitro and in vivo interaction using multiple complementary approaches, including Y2H, pull-down, co-immunoprecipitation (Co-IP), bimolecular fluorescence complementation (BiFC), and luciferase complementation imaging (LCI) assays (Fig. 3B–F). Among them, BiFC indicated that the interaction between SlSnRK2.6 and SlZHD8 occurs in the cell nucleus (Fig. 3E). The *Arabidopsis* genome contains 17 AtZHDs, whereas the tomato genome harbors 22 SlZHDs (Appendix Fig. S5F,G). While research on plant *ZHD* family has primarily focused on stress responses, their roles in sugar metabolism remain largely unexplored. Consequently, SlZHD8 represents a novel candidate that interacts with SlSnRK2.6 and functions in fruit sugar accumulation during salt stress.

## Root-derived ABA facilitates SlSnRK2.6-dependent phosphorylation of SlZHD8 during salt stress

We purified recombinant SlSnRK2.6-GST and SlZHD8-Strep proteins. Using Phos-Tag assay, we demonstrated that SlSnRK2.6 phosphorylates SlZHD8 in vitro (Fig. 3G). To identify the phosphorylation sites, we analyzed SlZHD8-Strep (phosphorylated by SlSnRK2.6-GST) using liquid chromatography-tandem mass spectrometry (LC-MS/MS). This revealed that threonine (T) at 247aa within the SlZHD8 DNA-binding domain is a likely phosphorylation site for SlSnRK2.6 (Appendix Fig. S6). We generated a non-phosphorylatable SlZHD8 variant (SlZHD8$^{T247A}$-Strep) by substituting T with alanine (A). In vitro kinase assay confirmed that SlZHD8$^{T247A}$-Strep showed abolished phosphorylation (Fig. 3G). To assess phosphorylation in vivo, we used the PVX system to express SlZHD8-Flag and SlZHD8$^{T247A}$-Flag in WT and *slsnrk2.6* fruits. Compared to WT, the phosphorylation level of SlZHD8-Flag in the *slsnrk2.6* background decreased by 57.55% (Fig. 3H). Furthermore, SlZHD8$^{T247A}$-Flag exhibited abolished phosphorylation in both genetic backgrounds (Fig. 3H).

We further utilized the PVX system to investigate the effects of salt stress and root-derived ABA on SlZHD8 phosphorylation in vivo. In WT fruits, exogenous ABA treatment enhanced SlZHD8 phosphorylation (Appendix Fig. S7A). However, this ABA-induced phosphorylation was almost abolished in *slsnrk2.6* mutant (Appendix Fig. S7A), indicating that ABA facilitates SlZHD8 phosphorylation through SlSnRK2.6. Next, we measured SlZHD8 phosphorylation levels in fruits from WT/WT and WT/*not* grafted plants under both control and salt stress conditions. Salt stress promoted SlZHD8 phosphorylation, but this promotion was reduced when root-derived ABA signaling was disrupted (in WT/*not*) (Fig. 3I). Furthermore, comparing SlZHD8 phosphorylation in WT and *slsnrk2.6* fruits under both control and salt stress conditions revealed that salt stress-induced phosphorylation of

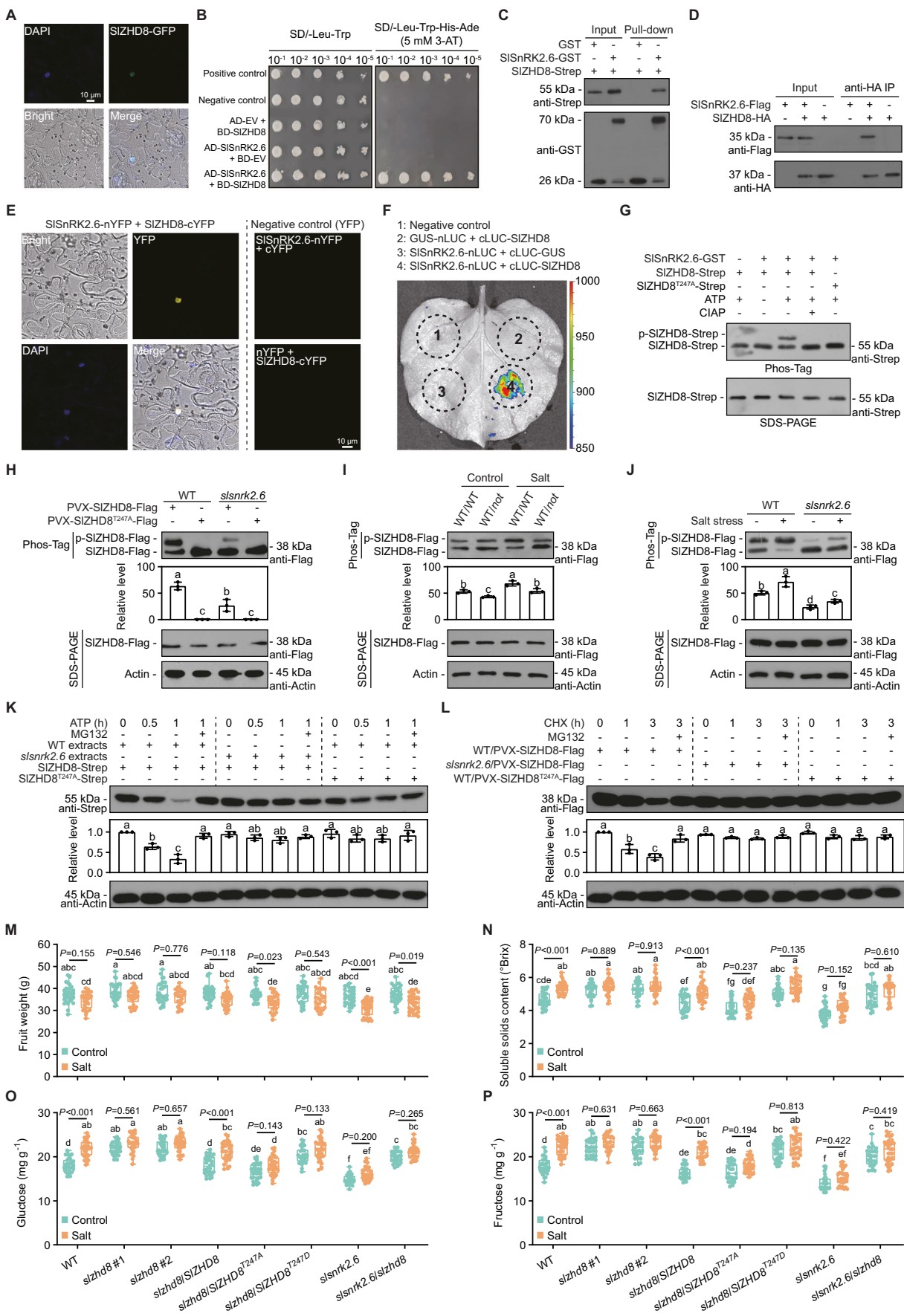

◄ **Figure 3. SlSnRK2.6 interacts with and phosphorylates SlZHD8 to mediate fruit sugar accumulation during salt stress.**

(A) Subcellular localization of SlZHD8. The SlZHD8-GFP and the nuclear localization marker DAPI were transiently co-infiltrated into *N. benthamiana* leaves for fluorescence observation. (B) Y2H assay shows that SlSnRK2.6 interacts with SlZHD8. The known interacting and non-interacting proteins were used as positive and negative control. (C) Pull-down assay shows that SlSnRK2.6 interacts with SlZHD8. Complex-bound proteins were analyzed by immunoblotting with anti-Strep and anti-GST antibodies, respectively. Empty GST protein was used as negative control. (D) Co-IP assay shows that SlSnRK2.6 interacts with SlZHD8. Constructs were co-expressed in WT leaves by PVX system for immunoblotting using anti-Flag and anti-HA antibodies, respectively. (E) BiFC assay shows that SlSnRK2.6 interacts with SlZHD8. Constructs were co-expressed in *N. benthamiana* leaves for fluorescence observation. The constructs with empty nYFP or cYFP were used as negative control. (F) LCI assay shows that SlSnRK2.6 interacts with SlZHD8. Constructs were co-expressed in *N. benthamiana* leaves. The known non-interacting proteins were used as negative control. (G) SlSnRK2.6 phosphorylates the T247 of SlZHD8 in vitro. Recombinant purified SlSnRK2.6-GST was incubated with SlZHD8-Strep or SlZHD8$^{T247A}$-Strep in kinase assay buffer. ATP was used to provide the phosphate group. Calf intestinal alkaline phosphatase (CIAP) was used for dephosphorylation. Protein abundance and mobility shift were detected by SDS-PAGE and Phos-Tag assay, respectively. (H) SlSnRK2.6 phosphorylates the T247 of SlZHD8 in vivo. Total proteins were extracted from PVX-mediated *SlZHD8-Flag* or *SlZHD8$^{T247A}$-Flag* transgenic fruits of WT and *slsnrk2.6* plants for phosphorylation assay. Protein abundance and mobility shift were detected by immunoblot and Phos-Tag assay, respectively. The ratio of p-SlZHD8-Flag is shown below the Phos-Tag image. Data represent mean ± SD ($n = 3$); statistical significance was determined by one-way ANOVA (Different letters indicate significant differences at $P < 0.05$). (I) Root-derived ABA is involved in the in vivo phosphorylation of SlZHD8 during salt stress. Total proteins were extracted from PVX-mediated *SlZHD8-Flag* transgenic fruits of WT/WT and WT/*not* plants in control and salt treatments for phosphorylation assay. Protein abundance and mobility shift were detected by immunoblot and Phos-Tag assay, respectively. The ratio of p-SlZHD8-Flag is shown below the Phos-Tag image. Data represent mean ± SD ($n = 3$); statistical significance was determined by one-way ANOVA (different letters indicate significant differences at $P < 0.05$). (J) SlSnRK2.6 is involved in the in vivo phosphorylation of SlZHD8 during salt stress. Total proteins were extracted from PVX-mediated *SlZHD8-Flag* transgenic fruits of WT and *slsnrk2.6* plants in control and salt treatments for phosphorylation assay. Protein abundance and mobility shift were detected by immunoblot and Phos-Tag assay, respectively. The ratio of p-SlZHD8-Flag is shown below the Phos-Tag image. Data represent mean ± SD ($n = 3$); statistical significance was determined by one-way ANOVA (Different letters indicate significant differences at $P < 0.05$). (K) SlZHD8 undergoes SlSnRK2.6-dependent degradation in cell-free degradation assay. The SlZHD8-Strep or SlZHD8$^{T247A}$-Strep was added into total protein extracts from WT and *slsnrk2.6* fruits. Reactions were incubated for indicated durations after supplement of 10 mM ATP. The MG132 was used as 26S proteasome inhibitor. Degradation was analyzed by immunoblotting using anti-Strep and anti-Actin (loading control) antibodies, respectively. Quantification of SlZHD8-Strep is shown below. Data represent mean ± SD ($n = 3$); statistical significance was determined by one-way ANOVA (Different letters indicate significant differences at $P < 0.05$). (L) SlZHD8 undergoes SlSnRK2.6-dependent degradation in vivo. Total proteins were extracted from PVX-mediated *SlZHD8-Flag* or *SlZHD8$^{T247A}$-Flag* transgenic fruits of WT and *slsnrk2.6* plants for degradation assay. The cycloheximide (CHX) and MG132 were used as protein synthesis inhibitor and 26S proteasome inhibitor, respectively. Degradation was analyzed by immunoblotting using anti-Flag and anti-Actin (loading control) antibodies, respectively. Quantification of SlZHD8-Flag was shown below. Data represent mean ± SD ($n = 3$); statistical significance was determined by one-way ANOVA (Different letters indicate significant differences at $P < 0.05$). (M–P) The *SlSnRK2.6-SlZHD8* module is essential for fruit sugar accumulation during salt stress. (M) Fruit weight, (N) soluble solids content, (O) glucose content, and (P) fructose content. Turning-stage fruits of indicated genotype in control and salt treatments were used for test. Box plots show median with 0.25 and 0.75 quartiles, whiskers represent values from minimum to maximum ($n = 30$); statistical significance was determined by one-way ANOVA (different letters indicate significant differences at $P < 0.05$). Source data are available online for this figure.

SlZHD8 is dependent on SlSnRK2.6 (Fig. 3J). Finally, using the T247A mutant, we confirmed that the phosphorylation of SlZHD8 under salt stress occurs primarily at T247 residue (Appendix Fig. S7B). Collectively, root-derived ABA facilitates SlSnRK2.6-dependent phosphorylation of SlZHD8 during salt stress.

## Phosphorylation of SlZHD8 reduces its protein stability

Gene expression of *SlZHD8* in fruits was unaffected by ripening, salt stress, or exogenous ABA treatment (Appendix Fig. S8A–C). However, SlZHD8 protein abundance significantly decreased during ripening (Appendix Fig. S8D), showing a negative correlation with SlSnRK2.6 kinase activity (Appendix Fig. S4I). Salt stress and ABA treatment both reduced SlZHD8 protein levels (Appendix Fig. S8E,F). So, we investigated whether SlSnRK2.6-mediated phosphorylation affects SlZHD8 stability. In cell-free degradation assay, SlZHD8-Strep was degraded slower in extracts from *slsnrk2.6* than in extracts from WT (Fig. 3K). Furthermore, SlZHD8$^{T247A}$-Strep exhibited greater stability than SlZHD8-Strep in WT extracts (Fig. 3K). To assess stability in vivo, we expressed SlZHD8-Flag or SlZHD8$^{T247A}$-Flag in WT and *slsnrk2.6* fruits using the PVX system. Following treatment with the protein biosynthesis inhibitor cycloheximide (CHX), SlZHD8-Flag degradation was monitored. Degradation was significantly slower in the *slsnrk2.6* background or when the T247A mutation was present (Fig. 3L). Moreover, degradation of SlZHD8 was blocked by the 26S proteasome inhibitor MG132 in both assays (Fig. 3K,L). Thus, SlSnRK2.6 phosphorylates SlZHD8 at T247, thereby reducing its stability.

## The *SlSnRK2.6-SlZHD8* module is essential for fruit sugar accumulation during salt stress

Using CRISPR-Cas9 technology, we generated two *slzhd8* mutants harboring a 1-bp deletion and a 46-bp insertion, respectively (Appendix Fig. S8G). Both mutations resulted in premature termination of the encoded protein. Knockout of *SlZHD8* alleviated salt stress-induced yield reduction (Fig. 3M), indicating that *SlZHD8* may negatively regulate salt tolerance. Compared to the WT, *slzhd8* accumulated higher levels of sugar in fruits under both control and salt stress conditions (Fig. 3N–P). Crucially, knockout of *SlZHD8* abolished the specific sugar accumulation induced by salt stress (Fig. 3N–P).

We generated three complementation lines in the *slzhd8* background, including WT complementation (*slzhd8/SlZHD8*), non-phosphorylatable complementation (*slzhd8/SlZHD8$^{T247A}$*), and phosphomimetic complementation (*slzhd8/SlZHD8$^{T247D}$*; substituting T with aspartic acid [D]). Under salt stress, fruit weight reduction was more pronounced in the non-phosphorylatable line compared to the WT and *slzhd8/SlZHD8* lines (Fig. 3M). By contrast, the phosphomimetic line showed no significant reduction in fruit weight (Fig. 3M). Both mutant lines abolished the salt stress-induced increase in fruit sugar accumulation observed in the WT and *slzhd8/SlZHD8* lines (Fig. 3N–P). However, sugar content was consistently lower in the non-phosphorylatable line than in the phosphomimetic line under both control and stress conditions (Fig. 3N–P). Notably, the sugar accumulation phenotype of the phosphomimetic line more closely resembled that of the *slzhd8* mutant (Fig. 3N–P).

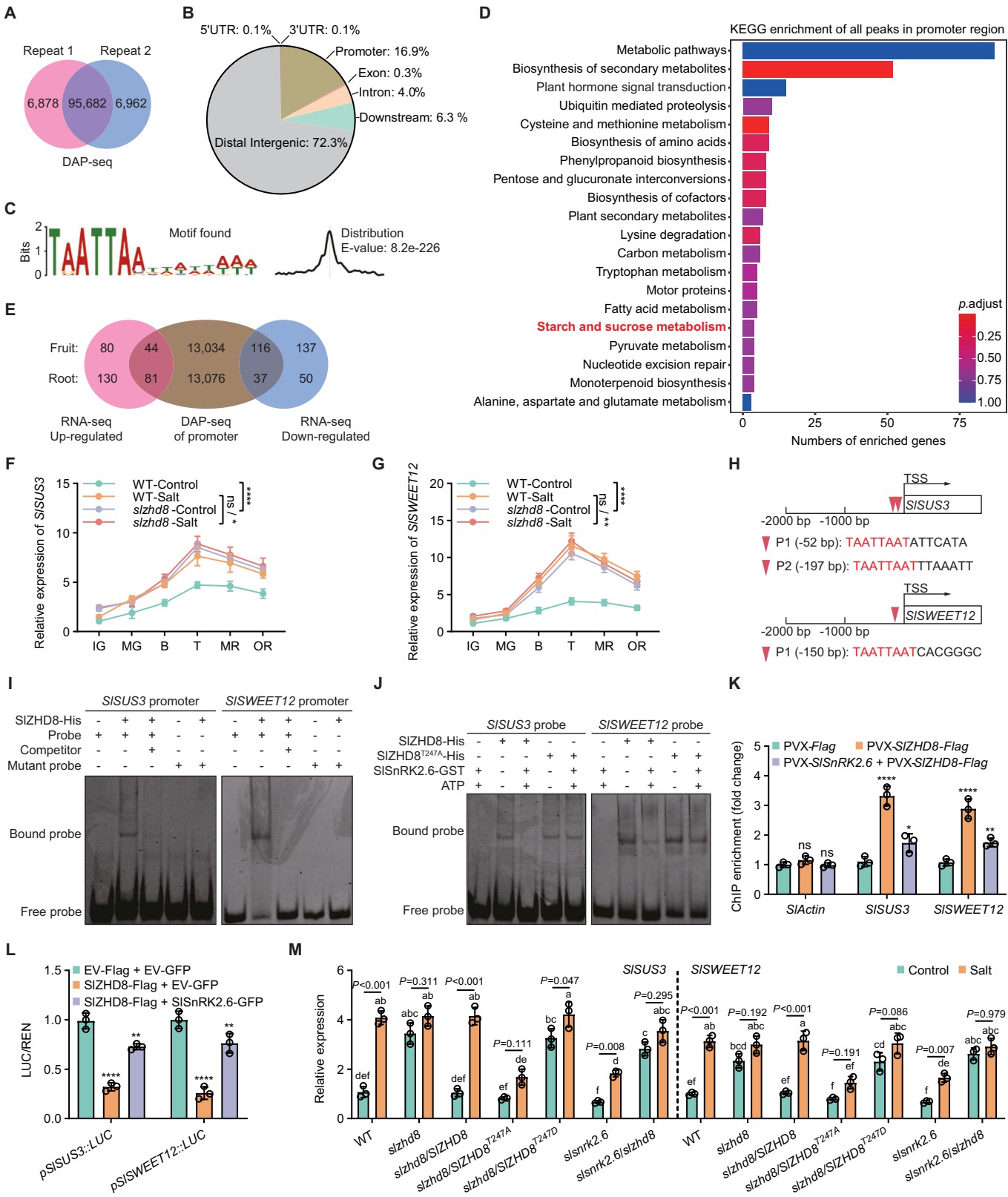

Figure 4. Phosphorylation of SlZHD8 by SlSnRK2.6 abolishes its transcriptional repression of *SlSUS3* and *SlSWEET12*.

(A) Venn diagram showing the peaks identified from DAP-seq. Refer to Dataset EV1. (B) Distribution of SlZHD8-binding peaks along the tomato genome. Refer to Dataset EV2. (C) The enriched SlZHD8-binding motifs within promoter regions. Refer to Dataset EV3. (D) KEGG enrichment analysis of SlZHD8-binding peaks within promoter regions. Significantly enriched KEGG terms ($P < 0.05$, hypergeometric test) were identified against the genomic background. Refer to Dataset EV5. (E) Venn diagram showing the extent of overlap between RNA-seq and DAP-seq. Refer to Dataset EV8. (F, G) Expression patterns of *SlSUS3* and *SlSWEET12* in WT and *slzhd8* fruits under control and salt stress. Fruit ripening: immature green, IG; mature green, MG; breaker, B; turning, T; mature red, MR; over red, OR. Data represent mean ± SD ($n = 3$); statistical significance was determined by two-way ANOVA ($^{ns}P > 0.05$, $*P < 0.05$, $**P < 0.01$, $****P < 0.0001$). (H) SlZHD8-binding motif in *SlSUS3* and *SlSWEET12* promoters. (I) EMSA shows direct binding of SlZHD8 to the *SlSUS3* and *SlSWEET12* promoter probes. Recombinant SlZHD8-His protein was incubated with 5′-FAM-labeled promoter probes. Negative controls included probe alone, addition of unlabeled probe (competitor), and mutant probe. (J) Phosphorylation of SlZHD8 by SlSnRK2.6 reduces DNA-binding capacity in EMSA. Recombinant SlZHD8-His and SlZHD8$^{T247A}$-His proteins were incubated either alone or with SlSnRK2.6-GST in kinase assay buffer; ATP was added to initiate phosphorylation. The resulting reaction mixtures were then used for EMSA as described previously. Mixtures lacking either SlZHD8-His or SlZHD8$^{T247A}$-His served as negative controls. (K) Phosphorylation of SlZHD8 by SlSnRK2.6 reduces its DNA-binding capacity in ChIP assay. Fruit samples expressing the indicated genes via the PVX system were harvested for ChIP. Anti-Flag antibody was used to immunoprecipitate the SlZHD8-DNA complex from the following samples: PVX-*Flag* (empty vector control), PVX-*SlZHD8-Flag* (SlZHD8-Flag expressed alone), and co-expressing PVX-*SlSnRK2.6* + PVX-*SlZHD8-Flag*. *SlActin* served as negative control. Data represent mean ± SD ($n = 3$); statistical significance was determined by two-way ANOVA ($^{ns}P > 0.05$, $*P < 0.05$, $**P < 0.01$, $****P < 0.0001$). (L) Phosphorylation of SlZHD8 by SlSnRK2.6 relieves its transcription inhibition in transactivation assay. *N. benthamiana* leaves expressing the indicated effector and reporter genes were used for transactivation assay. Effector: EV-Flag + EV-GFP (empty vector control), SlZHD8-Flag + EV-GFP (SlZHD8-Flag expressed alone), co-expressing SlZHD8-Flag + SlSnRK2.6-GFP. Reporters: *LUC* gene driven by either the *SlSUS3* or *SlSWEET12* promoter (*pSlSUS3::LUC* or *pSlSWEET12::LUC*). Data represent mean ± SD ($n = 3$); statistical significance was determined by two-way ANOVA ($**P < 0.01$, $****P < 0.0001$). (M) Phosphorylation of SlZHD8 by SlSnRK2.6 relieves its transcriptional inhibition of *SlSUS3* and *SlSWEET12*. Turning-stage fruits of indicated genotype in control and salt stress were used for test. Data represent mean ± SD ($n = 3$); statistical significance was determined by one-way ANOVA (different letters indicate significant differences at $P < 0.05$). Source data are available online for this figure.

To determine the genetic epistatic relationship between *SlSnRK2.6* and *SlZHD8* in salt stress-induced fruit sugar accumulation, we generated *slsnrk2.6/slzhd8* double mutant and compared its phenotype to *slsnrk2.6* and *slzhd8* single mutants. The *slsnrk2.6/slzhd8* phenocopied *slsnrk2.6* for fruit weight reduction under salt stress, but resembled *slzhd8* for fruit sugar accumulation (Fig. 3M–P). These findings indicate that while *SlSnRK2.6* plays a predominant role in regulating salt tolerance as measured by fruit weight, *SlZHD8* acts epistatically to *SlSnRK2.6* in controlling stress-induced fruit sugar accumulation. Specifically, *SlZHD8* functions downstream of *SlSnRK2.6* in this pathway. To determine whether the phosphorylation of SlZHD8 at T247 by SlSnRK2.6 mediates salt stress-induced fruit sugar accumulation, we expressed PVX-*SlZHD8*, PVX-*SlZHD8$^{T247A}$*, and PVX-*SlZHD8$^{T247D}$* in the fruits of *slzhd8* and *slsnrk2.6/slzhd8* backgrounds. Both *SlSnRK2.6* and *SlZHD8$^{T247A/D}$* mutations abolished the salt stress-induced fruit sugar accumulation phenotype (Appendix Fig. S9). Collectively, SlSnRK2.6-mediated phosphorylation of SlZHD8 at T247 regulates fruit sugar accumulation during salt stress, with *SlZHD8* exhibiting epistasis over *SlSnRK2.6* for this trait.

## Genome-wide mapping the transcriptional network of SlZHD8

To identify potential SlZHD8-binding motifs in target promoters, we performed DNA affinity purification sequencing (DAP-seq) assay. A total of 95,682 overlapping peaks were identified in two independent replicates (Fig. 4A; Dataset EV1). Genomic distribution analysis revealed that 16.9% of all peaks reside within promoters (Fig. 4B; Dataset EV2). To characterize SlZHD8-binding sites within promoters, we calculated the distance between each peak summit and the nearest transcription start site (TSS). A histogram of these distances ( ± 2 kb around TSSs) showed strong enrichment in the proximal promoter region, peaking approximately 300 bp upstream of TSSs (Appendix Fig. S10A; Dataset EV2). The highest-ranked SlZHD8-binding motif (TAATTAAT) within promoter regions were identified based on motif enrichment scores (Fig. 4C; Dataset EV3). Gene Ontology

(GO; Appendix Fig. S10B–D; Dataset EV4) and Kyoto Encyclopedia of Genes and Genomes (KEGG; Fig. 4D; Dataset EV5) enrichment analyses of genes (harboring SlZHD8-binding motif within promoter) indicated that putative SlZHD8 targets are significantly enriched in sugar metabolism-related processes. This enrichment included starch and sucrose metabolism (KEGG) and response to sucrose (GO: Biological Process), as well as associated hydrolase activity (GO: Molecular Function) and potential localization to the vacuolar membrane (GO: Cellular Component).

To further investigate the functional link between SlZHD8-binding motifs and its transcriptional regulatory role, we performed RNA sequencing (RNA-seq) of fruit and root samples from WT and *slzhd8* plants. This aimed to identify potential SlZHD8 target genes involved in regulating fruit sugar accumulation and root salt tolerance. Knocking out *SlZHD8* resulted in 124 up-regulated and 253 down-regulated genes in fruits, and 211 up-regulated and 87 down-regulated genes in roots (Fig. 4E; Dataset EV6). KEGG analysis confirmed significant enrichment of the starch and sucrose metabolism pathway in both organs (Appendix Fig. S11; Dataset EV7). Furthermore, Venn diagram overlap analysis identified key candidate target genes supported by both RNA-seq (differentially expressed) and DAP-seq (bound by SlZHD8) (Fig. 4E; Dataset EV8).

## Screening SlZHD8-targeted genes *SlSUS3* and *SlSWEET12*

Two potential target genes, *SlSUS3* and *SlSWEET12*, were selected from these overlapping genes based on the following rationale: SUS is a key enzyme directly hydrolyzing sucrose into glucose and fructose, thereby enhancing sink strength of fruit (Stein and Granot, 2019). *SlSUS3*, exhibiting the highest expression level in mature tomatoes, plays a critical role in regulating sugar accumulation (Zhang et al, 2024). SWEET functions as sugar transporter, potentially facilitating fruit sugar accumulation (Jia et al, 2024). Furthermore, *Arabidopsis* AtSWEET12 promotes sucrose transport from leaf to root, enhancing root growth and drought tolerance (Chen et al, 2022). Salt stress inhibits root

growth, an important factor limiting tomato growth and yield (Gong et al, 2014; Wei et al, 2025). Consequently, *SlSWEET12* represents a strong candidate for involvement in both fruit sweetness and salt tolerance.

RNA-seq analysis revealed that *SlZHD8* knockout caused up-regulation of both *SlSUS3* and *SlSWEET12* in fruit (2.54-fold and 2.02-fold) and up-regulation of *SlSWEET12* in root (3.55-fold), respectively (Dataset EV8). This up-regulation aligns with the phenotype where *SlZHD8* negatively regulates sugar accumulation (Fig. 3N–P) and supports the inference that *SlSUS3* and *SlSWEET12* promote sugar accumulation. To validate the screening results, we analyzed *SlSUS3* and *SlSWEET12* expression in both WT and *slzhd8* fruits during ripening under control and salt stress conditions (Fig. 4F,G). The expression patterns of both genes mirrored the sugar accumulation curve (Fig. 1D–F), maintaining high levels from breaker to over red stages. Salt stress induced *SlSUS3* and *SlSWEET12* expression in WT fruits compared to unstressed control (Fig. 4F,G). While *SlZHD8* knockout elevated baseline expression of both genes under control treatment, it abolished their salt stress induction (Fig. 4F,G). These findings support the hypothesis that SlZHD8 regulates *SlSUS3* and *SlSWEET12* to control fruit sugar accumulation.

We identified SlZHD8-binding motifs within the promoters of *SlSUS3* and *SlSWEET12* (Fig. 4H) and designed probes containing these motifs for electrophoretic mobility shift assay (EMSA). Recombinant SlZHD8-His protein directly bound to the 5'-FAM-labeled probes of the *SlSUS3* and *SlSWEET12* promoters, with this binding being competitively inhibited by unlabeled probes and abolished by mutated versions of the binding motifs (Fig. 4I). To examine in vivo association, we performed chromatin immuno-precipitation (ChIP) assay on PVX-mediated transgenic fruits. ChIP-qPCR analysis demonstrated significant enrichment of the *SlSUS3* and *SlSWEET12* promoters by SlZHD8-Flag compared to the control, confirming their direct regulation by SlZHD8 (Fig. 4K). To determine transcriptional regulation, we conducted transactivation assay using the LUC reporter gene driven by either the *SlSUS3* or *SlSWEET12* promoter (*pSlSUS3::LUC* or *pSlSWEET12::LUC*). Co-infiltration of SlZHD8-Flag with the reporter constructs into *N. benthamiana* leaves significantly reduced LUC activity relative to the empty vector (EV-Flag) control (Fig. 4L). These data demonstrate that SlZHD8 directly binds the *SlSUS3* and *SlSWEET12* promoters and functions as a transcriptional repressor.

## Phosphorylation of SlZHD8 by SlSnRK2.6 abolishes its transcriptional repression of *SlSUS3* and *SlSWEET12*

Since SlSnRK2.6-mediated phosphorylation site (T247) of SlZHD8 resides within its DNA-binding domain (Appendix Fig. S6B), we hypothesized that this modification might influence SlZHD8's DNA-binding ability. EMSA analysis revealed that adding SlSnRK2.6-GST with ATP weakened the binding of SlZHD8-His to *SlSUS3* and *SlSWEET12* probes (Fig. 4J). In contrast, the DNA-binding ability of SlZHD8$^{T247A}$-His was unaffected (Fig. 4J). Consistent with this, ChIP assay demonstrated that co-expressing SlSnRK2.6 with SlZHD8 (PVX-SlSnRK2.6 + PVX-SlZHD8-Flag) reduced the in vivo binding of SlZHD8 to *SlSUS3* and *SlSWEET12* promoters compared to expressing SlZHD8 alone (PVX-SlZHD8-Flag) (Fig. 4K). Furthermore, transactivation assays showed that co-expressing SlZHD8 with SlSnRK2.6 reduced the repressive effect

of SlZHD8 on the transcription of *SlSUS3* and *SlSWEET12* (Fig. 4L). Finally, we examined the effect of SlSnRK2.6-mediated SlZHD8$^{T247}$ phosphorylation on *SlSUS3* and *SlSWEET12* expression in fruits. Overall, phosphomimetic (*slzhd8/SlZHD8$^{T247D}$*) and non-phosphorylatable (*slzhd8/SlZHD8$^{T247A}$*) mimicked the effects of *slzhd8* and *slsnrk2.6* mutants on target genes' expression, respectively (Fig. 4M). These results position *SlZHD8* genetically downstream of *SlSnRK2.6* in regulating target genes' expression.

## The *SlZHD8-SlSUS3/SlSWEET12* module is essential for fruit sugar accumulation during salt stress

As *SlSUS3* is a key gene of sugar accumulation in fruits (Zhang et al, 2024), we focused on defining the genetic relationship between *SlZHD8* and *SlSUS3*. We silenced *SlSUS3* in both WT and *slzhd8* backgrounds by VIGS approach (Appendix Fig. S12A). Compared to empty vector (WT/TRV-*EV* and *slzhd8*/TRV-*EV*) controls, *SlSUS3* silencing significantly reduced fruit sugar accumulation in both WT/TRV-*SlSUS3* and *slzhd8*/TRV-*SlSUS3* backgrounds (Fig. 5A–C). Similar to *slzhd8*/TRV-*EV*, the fruits of both WT/TRV-*SlSUS3* and *slzhd8*/TRV-*SlSUS3* failed to increase sugar content during salt stress (Fig. 5A–C). Given the predominance of glucose and fructose in tomato fruits, we assessed whether SlSWEET12 transports these hexoses. Using the hexose transporter deficient yeast strain EBY4000 (Lu et al, 2025), we confirmed that *SlSWEET12* expression restored glucose and fructose uptake (Appendix Fig. S12B). Following this functional validation, we generated two *slsweet12* mutants (1-bp deletion and 1-bp insertion; Appendix Fig. S12C) and a *slzhd8/slsweet12* double mutant. Both *slsweet12* mutants significantly reduced fruit sugar accumulation under control and salt stress conditions compared to WT (Fig. 5D–F). Critically, like *slsweet12* mutants, the *slzhd8/slsweet12* mutant failed to increase sugar accumulation during salt stress (Fig. 5D–F). Together, both *SlSUS3* and *SlSWEET12* act downstream of *SlZHD8* to mediate salt stress-induced fruit sugar accumulation.

## The *SlZHD8-SlSWEET12* module is also involved in salt stress tolerance

When assessing salt stress effects on fruit weight, we observed that *slzhd8* mutant increased fruit weight (Fig. 3M), while *slsweet12* mutant decreased it relative to WT (Appendix Fig. S12E), suggesting both genes contribute to salt stress tolerance. Within 24 h of 0.6% NaCl treatment, root *SlZHD8* exhibited a rapid negative transcriptional response to salt stress, whereas root *SlSWEET12* showed a positive response (Appendix Fig. S13A,B). We therefore examined physiological responses of WT, *slzhd8*, *slsweet12*, and *slzhd8/slsweet12* seedlings under 0.6% NaCl stress (Fig. 5G–K; Appendix Fig. S13C–H). Compared to WT, *slzhd8* enhanced salt tolerance whereas *slsweet12* reduced it, as evidenced by differences in seedling growth vigor, root/shoot ratio, Na$^+$/K$^+$ homeostasis (a key tolerance indicator; Wang et al, 2020), and cytomembrane integrity (assessed via electrolyte leakage and malonaldehyde accumulation). These physiological changes correlated with root soluble sugar content. Importantly, the *slzhd8/slsweet12* double mutant displayed the same salt-hypersensitive phenotype as *slsweet12*, demonstrating that *SlSWEET12* is epistatic to *SlZHD8* in regulating salt stress tolerance.

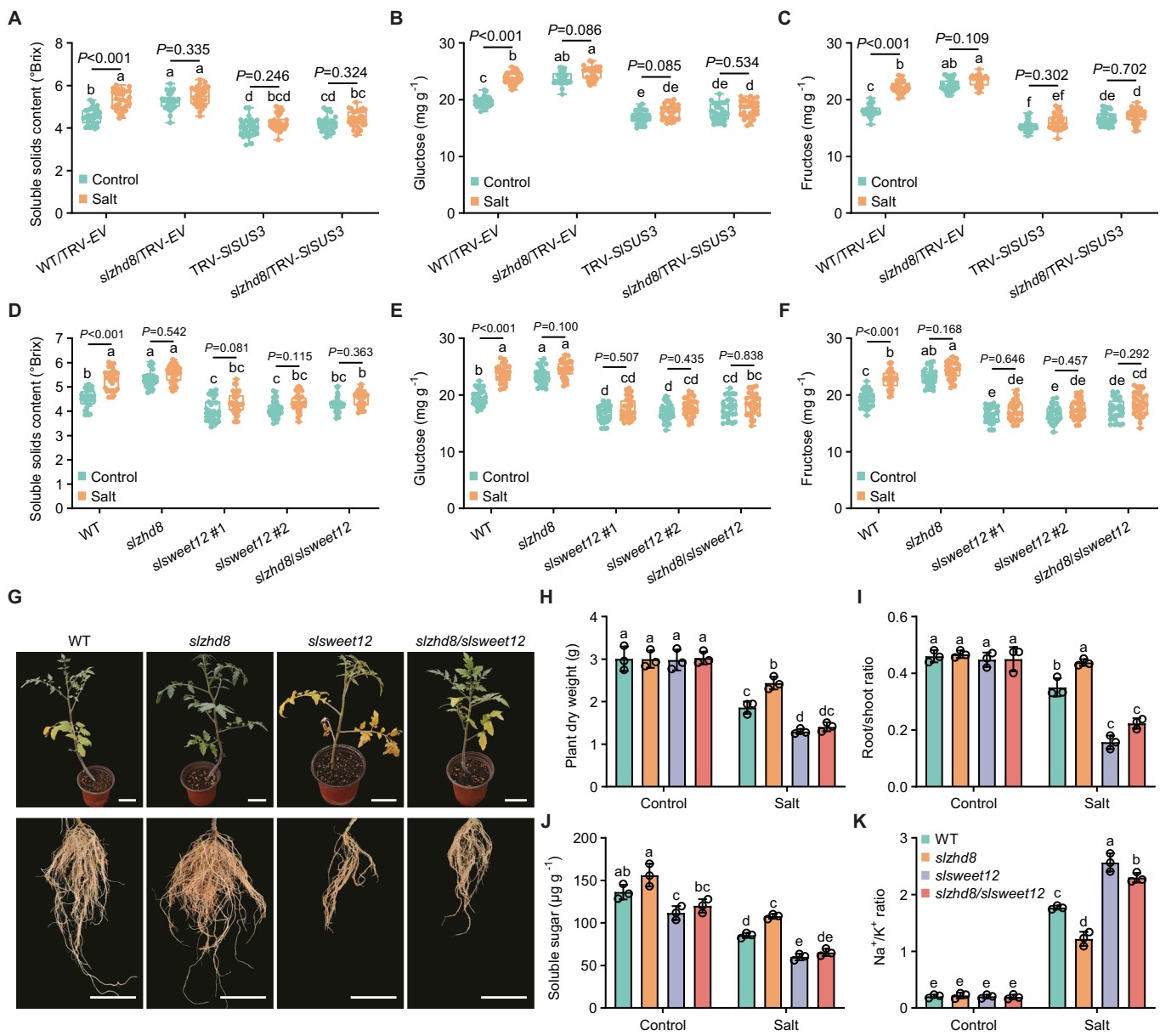

**Figure 5. The *SlZHD8-SlSUS3/SlSWEET12* module is involved in fruit sugar accumulation and stress tolerance during salt stress.**

(A–C) The *SlZHD8-SlSUS3* module is essential for fruit sugar accumulation during salt stress. (A) Soluble solids content, (B) glucose content, and (C) fructose content. Turning-stage fruits of TRV-EV (empty vector control) and TRV-SlSUS3 in both WT and *slzhd8* backgrounds under control and salt stress were used for test. Box plots show median with 0.25 and 0.75 quartiles, whiskers represent values from minimum to maximum ($n = 30$); statistical significance was determined by one-way ANOVA (Different letters indicate significant differences at $P < 0.05$). (D–F) The *SlZHD8-SlSWEET12* module is essential for fruit sugar accumulation during salt stress. (D) Soluble solids content, (E) glucose content, and (F) fructose content. Turning-stage fruits of WT, *slzhd8*, *slsweet12* (#1 and #2), and *slzhd8/slsweet12* under control and salt stress were used for test. Box plots show median with 0.25 and 0.75 quartiles, whiskers represent values from minimum to maximum ($n = 30$); statistical significance was determined by one-way ANOVA (Different letters indicate significant differences at $P < 0.05$). (G–K) The *SlZHD8-SlSWEET12* module is involved in salt stress tolerance. (G) Phenotypes of seedlings and roots under salt stress (scale bar = 5 cm), (H) plant dry weight, (I) root/shoot ratio, (J) soluble sugar content of root, and (K) $Na^+/K^+$ ratio of root. Seedlings of WT, *slzhd8*, *slsweet12*, and *slzhd8/slsweet12* were used for test at 20 d after salt stress (0.6% NaCl). Data represent mean ± SD ($n = 3$); statistical significance was determined by one-way ANOVA (different letters indicate significant differences at $P < 0.05$). Source data are available online for this figure.

## *ZHD8* is subject to artificial selection during tomato domestication

The domestication and improvement of tomato is hypothesized to have occurred in two major phases, as supported by genomic analyses (Razifard et al, 2020). The initial phase involved the transition from the wild species *S. pimpinellifolium* L. (SP) to the semi-domesticated *S. lycopersicum* L. var. *cerasiforme* (SLC) in South America. The second phase occurred in Mesoamerica, where SLC underwent further selection, leading to the fully domesticated *S. lycopersicum* L. var. *lycopersicum* (SLL). To identify genomic regions under artificial selection, we analyzed whole-genome

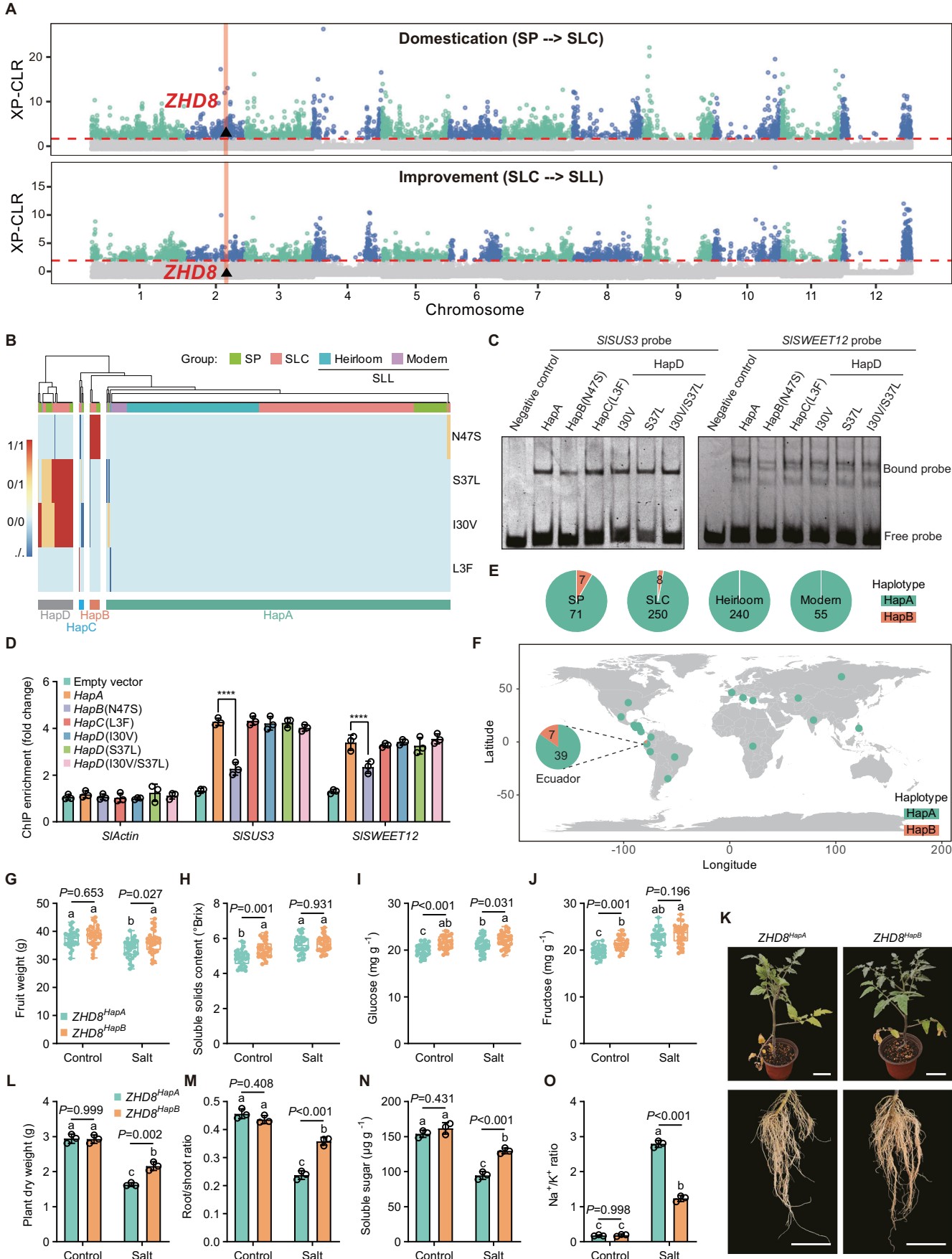

◀

**Figure 6. Natural variations of *ZHD8* are associated with fruit sugar accumulation and salt tolerance.**

(A) *ZHD8* is located within a domestication sweep but does not exceed the threshold for improvement sweep. Genome-wide detection of regions under artificial selection during tomato domestication and improvement. The red dotted line indicates the top 5% of regions. The black triangle represents the genomic position of *ZHD8*. (B) Haplotype clustering analysis of *ZHD8* across 631 tomato accessions, including SP, SLC, and SLL (heirlooms and modern cultivars). Haplotypes (Hap) A–D were identified through clustering analysis based on four missense variants (shown on right). (C) Haplotype ZHD8^HapB^ reduces the DNA-binding capacity in EMSA. Haplotypes of recombinant ZHD8^HapA^-His to ZHD8^HapD^-His proteins were incubated with 5′-FAM-labeled promoter probes of *SlSUS3* or *SlSWEET12*. Negative control included probe alone. (D) Haplotype ZHD8^HapB^ reduces the DNA-binding capacity in ChIP assay. Fruit samples expressing empty vector control and different *ZHD8* haplotypes via the PVX system were harvested for ChIP. Anti-Flag antibody was used to immunoprecipitate the ZHD8-DNA complex. Data represent mean ± SD ($n = 3$); statistical significance was determined by two-way ANOVA (****$P < 0.0001$). (E) Haplotype frequencies within tomato population groups of SP, SLC, and SLL (heirloom and modern cultivars). Numbers of *ZHD8^HapA^* and *ZHD8^HapB^* were shown in groups. (F) Geographic distribution of the *ZHD8^HapA^* and *ZHD8^HapB^*. Numbers of each haplotype were shown in population group of Ecuador. Geographic map was generated with the R package rworldmap. (G–J) Haplotype *ZHD8^HapB^* enhances fruit sugar accumulation during salt stress. (G) Fruit weight, (H) soluble solids content, (I) glucose content, and (J) fructose content. Turning-stage fruits of *ZHD8^HapA^* and *ZHD8^HapB^* under control and salt stress conditions were used for test. Box plots show median with 0.25 and 0.75 quartiles, whiskers represent values from minimum to maximum ($n = 30$); statistical significance was determined by one-way ANOVA (Different letters indicate significant differences at $P < 0.05$). (K–O) Haplotype *ZHD8^HapB^* enhances salt stress tolerance. (K) Phenotypes of seedlings and roots under salt stress (scale bar = 5 cm), (L) plant dry weight, (M) root/shoot ratio, (N) soluble sugar content of root, and (O) Na^+^/K^+^ ratio of root. Seedlings of *ZHD8^HapA^* and *ZHD8^HapB^* were used for test at 20 d after salt stress (0.6% NaCl). Data represent mean ± SD ($n = 3$); statistical significance was determined by one-way ANOVA (different letters indicate significant differences at $P < 0.05$). Source data are available online for this figure.

resequencing data from 631 accessions, including 78 SPs, 258 SLCs, and 295 SLLs (240 heirlooms and 55 modern cultivars) (Dataset EV9) (Zhou et al, 2022). Notably, *ZHD8* was identified as a candidate domestication-selected gene, specifically during the domestication phase but not the improvement phase, based on its presence within the top 5% of XP-CLR-ranked selective sweep regions (Fig. 6A).

Five natural variants of *ZHD8* have been identified in tomato populations. Four of these are missense mutations located within the gene's exonic regions, namely 2-37856316-T-C (asparagine to serine at 47aa; N47S), 2-37856346-G-A (serine to leucine at 37aa; S37L), 2-37856368-T-C (isoleucine to valine at 30aa; I30V), and 2-37856447-T-A (leucine to phenylalanine at 3aa; L3F) (Fig. 6B; Dataset EV9). To further explore functional variation at the *ZHD8* locus, we identified four major *ZHD8* haplotypes, each defined based on four missense variants (Fig. 6B). Based on the molecular function of *ZHD8*, we employed EMSA and ChIP assays to analyze the binding affinity of the four ZHD8 haplotypes to the promoters of *SlSUS3* and *SlSWEET12*. The results revealed that only the binding ability of ZHD8^HapB^ was significantly lower than that of ZHD8^HapA^ (Fig. 6C,D). Consequently, our subsequent studies primarily focused on the functional differences between *ZHD8^HapA^* and *ZHD8^HapB^*. *ZHD8^HapB^* was prevalent in South American accessions and exhibited a stepwise decline in frequency from SP to SLC to SLL (Fig. 6E,F). Remarkably, heirloom and modern cultivars exclusively harbor the *ZHD8^HapA^*, with no detectable presence of *ZHD8^HapB^* (Fig. 6E). This fixation indicates strong artificial selection for *ZHD8^HapA^* during domestication, resulting in its global predominance in cultivated tomatoes (Fig. 6F).

### *ZHD8^HapB^* enhances fruit sugar accumulation and salt tolerance in SLL

We expressed the *ZHD8^HapA^* (corresponding to the previously described *slzhd8/SlZHD8* line) and *ZHD8^HapB^* genes in the *slzhd8* mutant background and investigated the effects of these two haplotypes on fruit sugar accumulation and salt tolerance. Under control conditions, although *ZHD8^HapB^* had no significant effect on fruit weight, it significantly enhanced sugar accumulation in fruits (Fig. 6G–J). However, this promotive effect of *ZHD8^HapB^* on sugar accumulation was abolished under salt stress conditions (Fig. 6G–J).

Notably, using the same indicators employed in previous salt tolerance evaluations, we found that *ZHD8^HapB^* also conferred stronger salt tolerance to tomato seedlings (Fig. 6K–O; Appendix Fig. S14). This enhanced tolerance was further reflected in the differences in fruit weight observed under salt stress conditions (Fig. 6G).

## Discussion

The principle that "favorable conditions bring yield, stress enhances quality" is a well-known biological phenomenon frequently applied in agricultural production. However, the underlying scientific mechanisms remain insufficiently studied. This is challenging due to the complex interplay between plant genetics and environmental factors affecting crop quality, representing a significant gap in crop quality biology research. This study (summarized in Fig. 7) reveals: under salt stress, root-derived ABA acts as a key signal, transported to fruits to stimulate sugar accumulation. Then, elevated fruit ABA levels under salt stress activate SlSnRK2.6 kinase, which phosphorylates SlZHD8 at T247 residue; this phosphorylation inhibits SlZHD8 function by reducing its protein stability and DNA-binding activity, thereby lifting its transcriptional repression of *SlSUS3* and *SlSWEET12* to enhance fruit sugar accumulation. This study expands our understanding of the *SlSnRK2.6-SlZHD8-SlSWEET12* module, revealing its broader physiological roles in regulating root sugar accumulation and conferring salt tolerance. Investigation of *ZHD8* evolutionary patterns during tomato domestication revealed the beneficial haplotype *ZHD8^HapB^*, whose reduced binding affinity to target promoters enhances fruit sugar accumulation under non-stress conditions and enhances salt tolerance. These findings deepen our mechanistic understanding of "stress enhances quality" and highlight the potential of key genes, particularly *ZHD8* and its variants, in breeding tomatoes with enhanced sugar content and improved salt tolerance.

Unlike fruit, the root and leaf exhibit greater sensitivity to environmental fluctuations. These organs perceive changes in photoperiod, thermoperiod, soil moisture, and mineral nutrition, converting these environmental signals into genetic and chemical messages relayed to reproductive organs. This long-distance signaling systemically regulates plant reproductive timing and

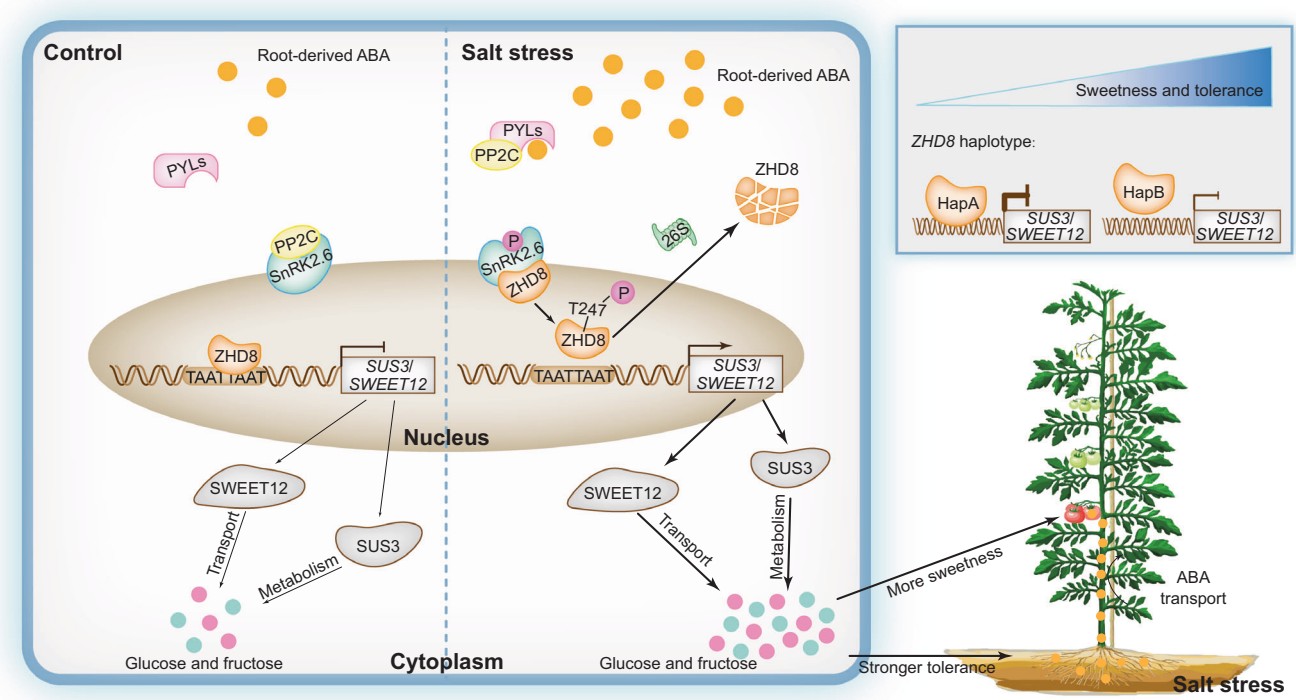

**Figure 7.  Mechanism model.**

Under salt stress, root-derived ABA transport to fruit was increased. Elevated fruit ABA activates SlSnRK2.6 kinase, which phosphorylates SlZHD8 and inhibits SlZHD8 function by reducing its protein stability and DNA-binding, thereby relieving its repression of *SlSUS3* and *SlSWEET12* to enhance fruit sugar accumulation. Meanwhile, the *SlSnRK2.6-SlZHD8-SlSWEET12* module also regulates root sugar accumulation and confers salt tolerance. Evolutionary analysis revealed a beneficial haplotype *ZHD8*[HapB]. Its reduced promoter-binding affinity promotes fruit sugar accumulation and salt tolerance. Source data are available online for this figure.

behavior, enabling adaptation to avoid adverse conditions. Vernalization and its associated semiochemicals exemplify plants' environmental adaptation strategy, utilizing long-distance signaling to sense changes and systemically coordinate growth and development (Xu and Chong, 2018). Specific instances of ABA signaling include root-derived ABA acting as a drought signal to promote early flowering in tomato (Chong et al, 2022) and leaf-derived ABA functioning as a temperature signal to regulate rice seed formation (Qin et al, 2021). These examples underscore the potential role of ABA long-distance transport in systemic perception and environmental adaptation. However, research attention in this field remains considerably less compared to systemic responses mediated by other hormones such as auxin, gibberellin, and jasmonic acid (Zandalinas et al, 2020). Reports specifically addressing ABA's role in transmitting soil salinity stress signals and regulating crop quality are even scarcer. While studies on crops like tomato (Tang et al, 2020) and strawberry (Perin et al, 2019) suggest that ABA metabolism induced by moderate salt or drought stress improves fruit quality, these primarily relied on omics analyses and lacked genetic validation to establish causal links between environment, hormone, and quality. Furthermore, the reasons and source of increased ABA levels in stressed fruit remain unclear, complicated by natural ABA accumulation during ripening (Zou et al, 2022). We demonstrate that salt stress triggers the transport and accumulation of root-derived ABA in fruits, a key physiological process for enhancing fruit sugar accumulation. The observation that root ABA is more sensitive to salt stress than leaf ABA is also shown in maize (Jia et al, 2002). This provides a physiological foundation for roles of root-derived ABA in regulating fruit energy metabolism via long-distance signaling. Nevertheless, it should be noted that the stress intensity induced by 0.1% NaCl is relatively mild. To clarify whether the *SlSnRK2.6-SlZHD8-SlSUS3/SlWEET12* pathway functions commonly across different salt stress conditions, we analyzed ABA content, SlSnRK2.6 kinase activity, SlZHD8 phosphorylation status, SlZHD8 protein stability, and transcript levels of *SlSUS3* and *SlWEET12* in fruits under 0.1–0.3% NaCl treatments (Appendix Fig. S1F–K). The results indicate that these physiological and biochemical responses follow consistent patterns across the tested salt concentrations, demonstrating that the core signaling pathway and mechanistic model proposed here are universal under varying saline conditions.

The initiation and amplification of SnRK2 activation are crucial events in ABA signaling (Lin et al, 2021). SnRK2s regulate fruit ripening and sugar accumulation in species like tomato (Zhu et al, 2023), watermelon (Wang et al, 2023), and apple (Jia et al, 2022). Furthermore, we demonstrated that salt stress further enhances ABA content and SlSnRK2.6 kinase activity in tomato fruit, suggesting that the ABA-SlSnRK2.6 signaling pathway regulates

fruit ripening and associated metabolic processes under salt stress. This functional link indirectly supports the rationale for targeting SlSnRK2.6. Given the dual role of SnRK2s in mediating plant root responses to salt stress (Lamers et al, 2025) and fruit ripening processes, we propose that the associated signaling pathway likely coordinates both stress tolerance and quality regulation. Recent studies show that the nitrate receptor NRT1.1B also functions as an ABA receptor, integrating nitrogen status and stress signals via competitive binding (Ma et al, 2025). This further reinforces the central role of ABA in regulating energy metabolism under stress conditions.

Currently, reports on plant *ZHD* transcription factors function primarily emphasize stress responses. While *SlZHD17*, which is a member of the same subfamily as *SlZHD8*, is the only reported *SlZHD* gene with a known function in tomato pigment metabolism (Shi et al, 2021), our work extends the functional scope of this subfamily. Beyond influencing fruit sugar metabolism, *slzhd8* mutants exhibited minimal impacts on agronomic traits under normal conditions but significantly enhanced performance under salt stress compared to WT (Appendix Fig. S15). This stress-specific enhancement, likely mediated by improved salt tolerance, highlights the breeding value of *SlZHD8*, as it improves stress resilience without growth penalties. We identified ZHD8 homologous proteins across 16 fruit-bearing crops examined, indicating that this transcription factor is conserved (Appendix Fig. S16A). The T247 site is conserved only in tomato and potato, indicating species-specificity in its phosphorylation pathway. In contrast, the adjacent T248 is highly conserved. Given their proximity and the overall conservation of this ZHD8 region, T248 may function as a phosphorylation site for SnRK2.6 or related kinases in other crops. These findings position ZHD8 as a promising target for breeding improved fruit sugar content.

To date, interactions between SnRK2s and ZHDs remain unreported in plants. While numerous studies have explored that ABA regulates soluble sugar accumulation to enhance osmotic tolerance, research on ABA signaling related to energy metabolism and sugar signaling predominantly focuses on SnRK1, TARGET OF RAPAMYCIN, and trehalose 6-phosphate (Morales-Herrera et al, 2024). We identified a novel sugar metabolism and transport signaling pathway regulated by the *SlSnRK2.6-SlZHD8-SlSUS3/ SlSWEET12* module. This pathway enhances fruit quality and salt tolerance by coordinately influencing sugar accumulation levels across different tissues. Analysis of gene expression patterns revealed that *SlSnRK2.6* is predominantly expressed in roots and fruits, whereas *SlZHD8* shows relatively low expression levels in these tissues (Appendix Fig. S16B). In contrast, *SlSWEET12* is highly expressed in leaves, with moderate expression in roots and fruits, and *SlSUS3* is mainly expressed in roots and fruits (Appendix Fig. S16B). These expression profiles support a model in which salt stress-induced ABA signaling, mediated by the *SlSnRK2.6-SlZHD8* module, promotes sugar loading into the phloem via SlSWEET12 for transport to storage organs like roots and fruits. Upon arrival, SlSUS3 mediates their metabolic conversion, and the resultant sugar accumulation in roots acts as an energy source and osmotic regulator, contributing to the common plant adaptation of salt stress resistance (Chen et al, 2022). However, the deeper biological significance of fruit quality regulation for tomato's adaptation to salt stress and reproductive success remains unknown. We propose that fruit sugar accumulation may not only promote energy storage

for seed maturation but also attract animal consumers, facilitating seed dispersal away from unfavorable environments. In *Arabidopsis*, AtSnRK2.2/2.3/2.6 phosphorylate AtSWEET11/12, promoting their oligomerization and sucrose transport activity (Chen et al, 2022). This phosphorylation facilitates sucrose transport to root, thereby improving root growth under drought stress. Our findings further elucidate the molecular mechanism whereby SlSnRK2.6 regulates *SlSWEET12* transcription through phosphorylation of SlZHD8. Additionally, we observed the significant contribution of *SlZHD8-SlSWEET12* module to promoting the root/shoot ratio under salt stress. Collectively, these results highlight the pivotal and intricate role of SnRK2s in plant adaptation to stress through the regulation of sugar transport.

Discovering and utilizing natural variation is fundamental to crop breeding. The *ZHD8^{HapB}* haplotype, which functionally resembles the phosphomimetic *SlZHD8^{T247D}* mutation, confers higher fruit sugar accumulation and stronger salt tolerance than *ZHD8^{HapA}*. Crucially, *ZHD8^{HapB}* directly enhances fruit sugar accumulation independently of salt stress. Furthermore, neither *slzhd8* mutants nor *ZHD8^{HapB}* haplotypes significantly alter core plant agronomic traits (Appendix Fig. S15). This functional stability enables diverse molecular modification strategies for *ZHD8*, highlighting its potential breeding value. Despite this potential, selection for *ZHD8^{HapB}* was overlooked during initial tomato domestication. Fortunately, this haplotype persists in some SLC varieties, enabling its rapid deployment in breeding programs. We also performed a genome-wide analysis of linkage disequilibrium (LD) decay in the tomato genetic population. The LD decay showed a clear gradient, being strongest in SLL, intermediate in SLC, and weakest in SP, which reflects the cumulative influence of breeding history. The high LD observed in SLL is consistent with historical directional selection and genetic drift during domestication (Appendix Fig. S17A; Lin et al, 2014). Furthermore, *ZHD8* was located outside the contiguous association block shared by neighboring loci, suggesting that it may function as an independent genetic determinant of the trait (Appendix Fig. S17B). Current research confirms that domestication has reduced tomato salt tolerance (Wang et al, 2020), explaining why the *ZHD8^{HapB}* to *ZHD8^{HapA}* transition was primarily driven by domestication rather than improvement selection.

## Conclusion

In tomato, root-derived ABA is transported to and accumulates within fruit during salt stress. The elevated ABA content in the fruit activates SlSnRK2.6 kinase to phosphorylate SlZHD8 at T247 residue, promoting SlZHD8 protein degradation and reducing its DNA-binding affinity. Consequently, the transcriptional repression of *SlSUS3* and *SlSWEET12* by SlZHD8 is relieved, leading to fruit sugar accumulation. Furthermore, the *SlSnRK2.6-SlZHD8-SlSWEET12* module also functions in roots, coordinating sugar accumulation and salt tolerance mechanisms under salt stress. Finally, population genetic analysis reveals that *ZHD8* underwent artificial selection during tomato domestication, yet the beneficial haplotype *ZHD8^{HapB}* has not been fixed in modern breeding lines. These converging insights, embedded within tomato domestication history, provide a molecular framework for understanding the adage "favorable conditions favor yield, while stress enhances quality." Importantly, we identify extant genetic variations that can

be harnessed for the synergistic improvement of both fruit quality and stress tolerance in breeding programs. We also acknowledge several limitations in this study. First, the broader applicability of the observed phenomena and related signaling pathways under a wider range of salt stress conditions or in natural saline environments remains to be verified. Furthermore, the robustness of our findings regarding fruit quality must be considered in light of the substantial variations among individual plants and across different ripening stages, necessitating a larger sample size to ensure reproducible results.

# Methods

### Reagents and tools table

| Reagent/resource | Reference or source | Identifier or catalog number |
|---|---|---|
| **Experimental models** | | |
| Ailsa Craig (*S. lycopersicum* L. var. *lycopersicum*) | Laboratory materials | N/A |
| LBA4404 (*Agrobacterium tumefaciens*) | Beyotime | Cat # D0393 |
| GV3101 (*A. tumefaciens*) | Beyotime | Cat # D0392 |
| **Recombinant DNA** | | |
| PGEX-6P-1 | Beyotime | Cat # BNCC360000 |
| pSmart VI | smart-lifesciences | Cat # SEC0100 |
| PET30(a)-His | Coolaber | Cat # VT063 |
| PGADT7 | Coolaber | Cat # VT001 |
| PGBKT7 | Coolaber | Cat # VT006 |
| pCAMBIA1300-cLuc | Novopro | Cat # V014944 |
| pCAMBIA1300-nLuc | Novopro | Cat # V014943 |
| pCBC-DT1T2 | Addgene | Cat # 50590 |
| pHSE401 | Addgene | Cat # 62201 |
| pSuper 1300+ | Novopro | Cat # V013425 |
| pGreenII 0800-LUC | Novopro | Cat # V010545 |
| pDR196 | Novopro | Cat # V001016 |
| **Antibodies** | | |
| BeyoGold™ GST-tag Purification Resin | Beyotime | Cat # P2251 |
| Strep-tag Purification Resin | Smart-Lifesciences | Cat # SA053025 |
| Anti-GST | Beijing Emarbio Science & Technology | Cat # EM34019 |
| Anti-Strep | Beyotime | Cat # AF2924 |
| Anti-HA Affinity Gel | Beyotime | Cat # P2287 |
| Anti-Flag | Beyotime | Cat # AG8050 |
| Anti-HA | Beyotime | Cat # AG8057 |
| Phos-Tag Acrylamide | Fujifilm wako chemicals | Cat # AAL-107 |
| Protein A Dynabeads | Life Technologies | Cat # 10006D |
| Anti-Actin | Sigma | Cat # A0840 |
| **Oligonucleotides and other sequence-based reagents** | | |

| Reagent/resource | Reference or source | Identifier or catalog number |
|---|---|---|
| PCR primers | This study | Dataset EV10 |
| **Chemicals, enzymes and other reagents** | | |
| ClonExpress II One Step Cloning Kit | Vazyme | Cat # C112-01 |
| $^3$H-ABA | American Radiolabeled Chemicals | Cat # ART1186 |
| RNA using the Fast King RT Kit | Vazyme | Cat # R312-01 |
| HiScript II Q RT SuperMix for qPCR | Vazyme | Cat # R223-01 |
| Yeast Protocols Handbook | Clontech | Cat # PT3024-1 |
| The 4′,6-diamidino-2-phenylindole | Solarbio | Cat # C0065 |
| DAP-seq Kit | Bluescape | Cat # D202009 |
| **Software** | | |
| VIGS tool | http://vigs.solgenomics.net | |
| CRISPOR tool | http://crispor.tefor.net | |
| Heinz 1706 genome | https://github.com/bwa-mem2/bwa-mem2 | |
| RNA-seq raw data | https://www.ncbi.nlm.nih.gov/sra | |
| **Other** | | |
| Illumina HiSeq 4000 | Illumina | |

## Gene accession numbers, vectors, and primers

Gene accession numbers, vectors, and primers are presented in Dataset EV10.

## Plant materials and genetic manipulations

Wild-type (WT) tomato material: *S. lycopersicum* L. var. *lycopersicum* 'Ailsa Craig'.

Mutants construction: The *slsnrk2.6*, *slzhd8*, and *slsweet12* mutants were generated using CRISPR-Cas9. Two sgRNAs targeting the coding sequences of target genes were designed using CRISPOR tool (http://crispor.tefor.net/), and then were cloned into vector pHSE401 via homologous recombination by ClonExpress II One Step Cloning Kit (Cat # C112-01, Vazyme, China). Constructs were introduced into *Agrobacterium tumefaciens* LBA4404 strains for stable transformation of WT plants. Mutants were confirmed by PCR and sequencing of genomic DNA. Homozygous were used for experiments. The *slsnrk2.6/slzhd8* and *slzhd8/slsweet12* double mutants were generated by crossing the corresponding single mutants. The SlZHD8^T247A, SlZHD8^T247D, and ZHD8^HapB genes were obtained by site-directed mutagenesis. *SlZHD8* (also *ZHD8^HapA*), *SlZHD8^T247A*, *SlZHD8^T247D*, or *ZHD8^HapB* was separately fused into the pCAMBIA1300 driven by the native *SlZHD8* promoter using *slzhd8* #1 lines as background for constructions of *slzhd8/SlZHD8* (also *ZHD8^HapA*), *slzhd8/SlZHD8^T247A*, *slzhd8/SlZHD8^T247D*, and

$ZHD8^{HapB}$ via LBA4404-mediated genetic transformation. The *not* (*notabilis*, 'Ailsa Craig' background) mutant that has *SlNCED1* mutation and ABA-deficient phenotype was obtained from the Tomato Genetics Resource Center.

Virus-induced gene silencing (VIGS): VIGS was performed using *Tobacco rattle virus* (TRV) vectors TRV1 and TRV2 (Wei et al, 2025). Target-specific sequences of target genes were designed by VIGS tool (vigs.solgenomics.net), and then were amplified and cloned into TRV2, respectively. TRV1 and TRV2 constructs in LBA4404 strains were used to co-infect fruits or seedlings as previously described (Liu et al, 2024a). TRV empty vector was used as the control. Plant samples need to undergo gene expression testing before experiments.

Virus-mediated gene expression: Gene expression was performed using *Potato virus X* (PVX) vectors (Wei et al, 2025). The coding sequences of target genes were amplified and cloned into PVX under the CaMV35S promoter with a Flag or HA tag. PVX empty vector was served as the control. Reconstructed vectors in LBA4404 strains were used to infect plant samples via a similar inoculation method as VIGS. Plant samples need to undergo gene expression testing before experiments.

## Salt stress treatment

All experiments were carried in a solar greenhouse (Taian, China).

Fruit quality experiment: Thirty-day-old seedlings of specified tomato genotypes were cultivated in 15 L containers (diameter 45 cm, height 40 cm) with normal field soil. Upon reaching the flowering stage, tomato plants were irrigated weekly either with water (Control) or with 0.1% NaCl solution (Salt stress). Each irrigation was applied until the soil was fully saturated. This method ensures that excess NaCl solution drains through the bottom of the container, thereby maintaining a consistent soil NaCl concentration of ~0.1%. The salt stress treatment was maintained throughout the entire growth cycle of the tomatoes. Each treatment contained at least 60 plants. In fruit weight and sugar assays, every five fruits of each plant were regarded as a single biological replicate, and a total of thirty biological replicates were set up. In ABA and gene expression assays, samples from every ten plants were mixed into one sample for a single biological replicate, and a total of three biological replicates were set up.

Seedling tolerance experiment: Tomato seedlings of the specified genotypes were pot-grown using a commercial potting mix. At 30 days old, the plants were irrigated daily with either water (Control) or 0.6% NaCl solution (Salt stress). Each watering was applied until the soil reached full saturation to ensure a stable NaCl concentration in the medium. The salt stress treatment was continued for a period of 20 days. Each treatment contained at least 50 plants. Every ten seedlings were mixed into one sample for a single biological replicate, and a total of three biological replicates were set up.

## Fruit weight and sugar assays

The ripening process of tomatoes can be divided into six stages from 32 days post anthesis (dpa) to 60 dpa, including immature green (IG), mature green (MG), breaker (B), turning (T), mature red (MR), and over red (OR) (Fig. 1A). After the initial clarification of the effects of salt stress on the fruit quality at different ripening

process (Fig. 1A–G), subsequent studies all used the turning-stage fruits for testing. Five fruits of each plant were mixed into one sample for assaying the contents of soluble solids, glucose, and fructose. The soluble solids content was determined using a digital refractometer, adjusted and calibrated at 20 °C with distilled water and expressed as degrees °Brix. Glucose and fructose contents were quantified using a Waters e2695 High-Performance Liquid Chromatography (HPLC) system (Waters Alliance, USA), equipped with a Waters 2414 refractive index detector and a Sugar-Pak II column (Lu et al, 2025). The temperatures of the column and refractive index detector were set at 80 °C and 35 °C, respectively. The injection volume was 10 μL, and ultrapure water at a flow rate of 0.6 mL min$^{-1}$ was used as the mobile phase. All testing were conducted with thirty biological replicates.

## ABA measurement

According to Qin et al, (2021), plant samples (50 mg) were ground to powder in liquid nitrogen, dissolved in 1 mL of ethyl acetate with 10 μL of internal standard (600 ng mL$^{-1}$ chloromycetin), and then incubated with shaking overnight at 4 °C. Subsequently, the samples were centrifuged for 15 min (15,000× $g$, 4 °C) to obtain the supernatants. The pellets were reextracted by adding 0.5 ml ethyl acetate without internal standards and incubating for 5 h at 4 °C in the dark and then centrifuged for 15 min (15,000× $g$, 4 °C) to obtain the supernatants. The supernatants of two extractions were combined and dried using a Termovap sample concentrator. The dried extracts were dissolved in 200 μL of 70% methanol, vortexed for 25 min, and then centrifuged for 10 min (15,000× $g$, 4 °C). The supernatants were respectively transferred to 1.5-mL LC vials and then injected into the HPLC (1100, Agilent, USA) coupled to a triple four-pole tandem mass spectrometer (MS) (API3000, AB Science, USA) for ABA quantification. For HPLC, the chromatographic separation was performed on an Inertsil ODS-3 column (2.1 mm by 100 mm, 5 μm), protected by a Phenomenex C18 guard column (4 mm by 3 mm inner diameter; Torrance, CA, USA). The temperature of column was 30 °C. The mobile phases and gradient elution are as follows: solvent A (0.1% formic acid), 45–65% for 0 to 5 min and 65–45% for 5 to 6 min; solvent B (methanol), 55–35% for 0 to 5 min and 35–55% for 5 to 6 min and then equilibrated for 2 min in 45% solvent A and 55% solvent B. The injection volume and flow rate were 10 and 0.3 mL min$^{-1}$, respectively. The retention times were 4.53 min for ABA and 3.23 min for chloromycetin. MS was performed in the negative ion mode with electrospray ionization tandem MS. The main working parameters of MS were optimized as follows: nebulizer gas, 8 L·min$^{-1}$; curtain gas, 10 L·min$^{-1}$; collision activated dissociation gas, 4 L·min$^{-1}$; turbo ion spray voltage IS, -4000 V; and source temperature TEM, 450 °C. The compound parameters such as declustering potential, focusing potential, entrance potential, collision energy, and collision cell exit potential were optimized and set to −57.58, −157.3, −5.97, −18.91, and −16.9 for ABA and to −52.96, −198, −9.2, −25.75, and −6.87 for chloromycetin, respectively. Analytes were detected by tandem MS using multiple reaction monitoring of the most intensive precursor-fragment transitions with 200-ms dwell time, at a mass/charge ratio (*m/z*) of 263.1/152.9 for ABA and *m/z* of 321.0/151.8 for chloromycetin. The data were analyzed using Analyst 1.4.1 software. ABA measurement was performed with three biological replicates.

## ³H-ABA feeding experiment

The dwarfing tomato variety 'Micro-Tom' was used for ³H-ABA (Cat # ART1186, American Radiolabeled Chemicals) feeding experiments. Root or leaf tissues were inserted into 10 mL of water with 5 μL of ³H-ABA. At 6 and 12 h after feeding, Fruit samples were ground and then mixed with 5 mL of scintillation solution for monitoring radioactivity using Tri-Carb 2910TR Low Activity Liquid Scintillation Analyzer (PerkinElmer, USA). Feeding assay was conducted with three biological replicates.

## Salt tolerance assessment

Every ten seedlings were mixed into one sample for a single biological replicate, and a total of three biological replicates were set up in salt tolerance assessment, including phenotype, plant dry weight, root/shoot ratio, root $Na^+$ content, root $K^+$ content, root $Na^+/K^+$ ratio, root soluble sugar content, root electrolyte leakage, and root malonaldehyde content. Phenotype was obtained by camera. Dry weight of shoot and root was measured using an electronic balance, and the root/shoot ratio was calculated then. Electrolyte leakage of root was measured using a conductometer (DDSJ-308F, Leici, China). The malonaldehyde, $Na^+$, $K^+$ contents, and $Na^+/K^+$ ratio were measured according to our previous description (Wei et al, 2025). Soluble sugar content was determined by the anthrone colorimetry method (Liu et al, 2024c).

## Gene expression analysis

Gene expression was quantified by real-time quantitative PCR (RT-qPCR) approach following MIQE guidelines (Bustin et al, 2025), using *SlActin* as the reference gene. Total RNA was extracted using the TRIzol method. cDNA was synthesized from RNA using the Fast King RT Kit (Cat # R312-01, Vazyme, China). RT-qPCR was performed using HiScript II Q RT SuperMix for qPCR (Cat # R223-01, Vazyme, China) in 25 μL reactions containing 800 ng cDNA and 100 nM primers. Relative gene expression was calculated using the $2^{-\Delta\Delta Ct}$ method (Liu et al, 2023).

## Protein extraction and immunoblotting assay

Plant samples (50 mg) were homogenized in 150 μL extraction buffer (100 mM Tris-HCl pH 7.4, 100 mM NaCl, 5 mM EDTA, 5% SDS, 20% glycerol, 20 mM DTT, 40 mM β-mercaptoethanol, 2 mM PMSF, 1× protease inhibitor cocktail, 80 μM MG132, 80 μM MG115, 1% phosphatase inhibitor cocktail, 10 mM *N*-ethylmaleimide). Samples were boiled for 10 min, then centrifuged at 16,000 *g* for 10 min. Supernatants were resolved by SDS-PAGE and transferred to PVDF membranes. Membranes were blocked with 5% (w/v) non-fat milk. Target proteins were detected using specific primary antibodies followed by labeled secondary antibodies. Protein bands were visualized by chemiluminescence and quantified using ImageJ.

## Yeast two-hybrid (Y2H) assay

Y2H assay was performed using pGADT7 (AD, prey) and pGBKT7 (BD, bait) vectors according to the Yeast Protocols Handbook (Cat # PT3024-1, Clontech, USA). The screen for prey proteins interacting with bait proteins utilized a tomato cDNA library. Point-to-point protein interaction verification was subsequently carried out. Prey (AD-SlSnRK2.6) and bait (BD-SlZHD8) vectors were constructed and co-transformed into the Y2H Gold yeast strain. Known interacting proteins (SV40 large T antigen and p53; AD-T + BD-53) and non-interacting proteins (SV40 large T antigen and Lamin C; AD-T + BD-Lam) served as positive and negative controls, respectively (Cho et al, 1994). Transformants were plated on SD/-Leu-Trp medium to select for colonies containing both plasmids. Single colonies from SD/-Leu-Trp were then replica-plated onto SD/-Leu-Trp-His-Ade with 5 mM 3-AT medium to select for interactions between prey and bait proteins.

## Luciferase complementation imaging (LCI) assay

The coding sequences of *SlSnRK2.6* and *SlZHD8* were separately cloned into nLUC (N-terminal fragment) and cLUC (C-terminal fragment) vectors. These constructs were co-expressed in *N. benthamiana* leaves by infiltration of *A. tumefaciens* GV3101 strains. Luciferase (LUC) activity was detected using a IVIS Lumina In Vivo Imaging System (PerkinElmer, USA) at 2 dpi. Known non-interacting proteins (p22 and NbBAG1) served as the negative control (Shang et al, 2023).

## Bimolecular fluorescence complementation (BiFC) assay

The coding sequences of *SlSnRK2.6* and *SlZHD8* were separately cloned into vectors containing the N-terminal (nYFP) or C-terminal (cYFP) fragments of YFP, generating SlSnRK2.6-nYFP and SlZHD8-cYFP constructs. These constructs were co-expressed in *N. benthamiana* leaves via GV3101-mediated infiltration. YFP fluorescence complementation in *N. benthamiana* leaves was imaged at 3 dpi using a Zeiss LSM 880 confocal laser scanning microscope (CLSM). YFP was excited at 514 nm and emission was captured between 525 and 585 nm.

## Pull-down assay

SlSnRK2.6-GST and SlZHD8-Strep recombinant proteins were purified using BeyoGold™ GST-tag Purification Resin (Cat # P2251, Beyotime, China) and Strep-tag Purification Resin (Cat # SA053025, Smart-Lifesciences, China), respectively. For pull-down assay, SlSnRK2.6-GST bound to resin was incubated with SlZHD8-Strep in binding buffer (50 mM Tris-HCl pH 8.0, 200 mM NaCl, 1 mM EDTA, 1% NP-40, 1 mM DTT, 10 mM $MgCl_2$) at 4 °C for 2–3 h with gentle rotation. The resin was pelleted by centrifugation at 3000×*g* for 5 min and washed three times with wash buffer (50 mM Tris-HCl pH 8.0, 400 mM NaCl, 1 mM EDTA, 1 mM DTT). Bound protein complexes were eluted with elution buffer (50 mM Tris-HCl pH 8.0, 20 mM reduced glutathione). Eluted proteins were detected by immunoblotting using anti-GST (Cat # EM34019, Beijing Emarbio Science & Technology, China) and anti-Strep (Cat # AF2924, Beyotime, China) antibodies.

## Co-immunoprecipitation (Co-IP) assay

PVX-based vectors harboring *SlSnRK2.6* and *SlZHD8* were constructed, respectively. Expression of SlSnRK2.6-Flag and SlZHD8-HA proteins in tomato plants was mediated by PVX as

described previously. Plant samples were lysed in extraction buffer (10 mM Tris-HCl pH 7.6, 0.5% Nonidet P-40, 2 mM EDTA, 150 mM NaCl, 1× EDTA-free protease inhibitor cocktail [Roche]). Lysates were centrifuged and the supernatant was incubated with pre-washed anti-HA Affinity Gel (Cat # P2287, Beyotime, China) for 2 h at 4 °C. Following five washes, co-immunoprecipitated proteins were detected via immunoblotting with anti-Flag (Cat # AG8050, Beyotime, China) and anti-HA (Cat # AG8057, Beyotime, China) antibodies.

## Electrophoretic mobility shift assay (EMSA)

EMSA was performed according to Liu et al (2024b). The coding sequence of *SlZHD8* (or its variant) was cloned into pET30a vector. SlZHD8-His recombinant protein (or its variant) was expressed in *E. coli* following induction with 1 mM IPTG at 28 °C for 10 h. The DNA-binding motif fragments derived from promoters of target genes were synthesized and 5'-end labeled with 5-FAM. For binding reactions, 0.08 pmol of labeled probe was incubated with 2.5 μg of purified proteins in 20 μL binding buffer (20 mM Tris-HCl pH 7.5, 100 mM NaCl, 2 mM MgCl$_2$, 1 mM DTT, 10% glycerol) at 25 °C for 30 min. Negative control consisted of probe alone. For competition assay, unlabeled competitor DNA was added in tenfold molar excess. Protein-DNA complexes were resolved on native 8% polyacrylamide gels in 0.5× TBE at 140 V for 90 min (4 °C, dark). Gels were imaged using a Tanon 5200 Multi system. For kinase-mediated EMSA assay, recombinant SlZHD8-His protein (or its variant) was incubated with SlSnRK2.6-GST in kinase assay buffer (20 mM Tris-HCl buffer, 100 mM NaCl, 20 mM MgCl$_2$, 2 mM DTT, 10 mM of ATP) at 30 °C for 30 min. After the reactions, the mixtures were used for EMSA as previous.

## Chromatin immunoprecipitation qPCR (ChIP-qPCR) assay

PVX-*Flag* (negative control) and PVX-*SlZHD8-Flag* (or its variant) vectors were constructed and expressed in tomato plants. ChIP assay followed established methods (Du et al, 2017). Briefly, 5 g of samples was vacuum-infiltrated with 1% formaldehyde for 10 min, flash-frozen in liquid nitrogen, and ground to a fine powder. Chromatin was isolated, sonicated, and immunoprecipitated with anti-Flag antibody (Cat # AG8050, Beyotime, China). Purified ChIP DNA and input DNA were resuspended in 30 μL nuclease-free water. qPCR signals were normalized to input DNA, with fold enrichment calculated relative to the *SlActin* promoter.

## Transactivation assay

The *SlZHD8* coding sequence was cloned into the pGreen II 0029 62-SK effector vector, and the *SlSUS3* or *SlSWEET12* promoter was inserted into the pGreen II 0800-LUC reporter vector (Shan et al, 2025). Constructs were electroporated into GV3101. *N. benthamiana* leaves were infiltrated with transformed GV3101, and Firefly luciferase (LUC) and Renilla luciferase (REN) activities were measured at 3 dpi. Luminescence was quantified using a Synergy MX full wavelength multifunctional microplate reader (BioTek, USA). The LUC/REN ratio for the empty effector vector plus promoter construct was set to 1 for normalization.

## In vitro phosphorylation and LC-MS/MS assays

In vitro phosphorylation assay was performed in kinase assay buffer (20 mM Tris-HCl buffer, 100 mM NaCl, 20 mM MgCl$_2$, 2 mM DTT, 10 mM ATP) using GST-fused and Strep-fused proteins. The samples were incubated for 30 min at 30 °C, and the reactions were stopped by adding 5× loading buffer and boiling for 5 min. Then, the samples were separated using 10% SDS-PAGE, with 0.1 mM of MnCl$_2$ and 0.1 mM of Phos-Tag Acrylamide (Cat # AAL-107, Fujifilm Wako Chemicals, Japan) and the proteins were detected using an anti-Strep antibody (Cat # AF2924, Beyotime, China). For LC-MS/MS assay, phosphorylated SlZHD8-Strep were obtained by in vitro phosphorylation. Proteins were separated by 10% SDS-PAGE. After Coomassie Brilliant Blue staining, the corresponding band was cut for LC-MS/MS analysis.

## In vivo phosphorylation assay

To study the in vivo phosphorylation of SlZHD8, total protein was extracted from PVX-mediated transgenic samples with protein extraction buffer (50 mM Tris-HCl, pH 7.4, 150 mM NaCl, 2 mM MgCl$_2$, 20% glycerol, 5 mM DTT, 0.1% Nonidet P-40) containing protease inhibitor cocktail. Sample cell debris was pelleted, and the supernatant was incubated with anti-Flag antibody (Cat # AG8050, Beyotime, China) at 4 °C overnight to capture the epitope-tagged protein. On the second day, 50 mL of Protein A Dynabeads (Cat # 10006D, Life Technologies, USA) was added. After 4 h of incubation at 4 °C, the beads were collected and washed three times with washing buffer (50 mM Tris-HCl, pH 7.4, 150 mM NaCl, 2 mM MgCl$_2$, 10% glycerol, 5 mM DTT, and 0.1% Nonidet P-40). The immunoprecipitated proteins were released from the beads by boiling in sample loading buffer and analyzed by immunoblotting with anti-Flag in 12% Phos-Tag SDS-PAGE as previous in vitro phosphorylation.

## Protein degradation assay

For cell-free protein degradation assay, total protein was extracted from WT and *slsnrk2.6* fruits. Equal amounts of protein extract were incubated with purified substrate proteins of SlZHD8-Strep or SlZHD8$^{T247A}$-Strep and 10 mM ATP for designated time periods. MG132 (50 μM) was added to inhibit the 26S proteasome. Proteins were separated by SDS-PAGE and immunoblotted with anti-Strep (Cat # AF2924, Beyotime, China) and anti-Actin (Cat # A0840, Sigma, USA) antibodies.

For PVX-mediated protein degradation assay, the constructed PVX-SlZHD8-Flag or PVX-SlZHD8$^{T247A}$-Flag was separately injected into WT and *slsnrk2.6* fruits, resulting in three genotypes of WT/PVX-SlZHD8-Flag, *slsnrk2.6*/PVX-SlZHD8-Flag, and WT/PVX-SlZHD8$^{T247A}$-Flag. These fruits were treated with only 200 μM CHX (protein synthesis inhibitor) or 200 μM CHX + 50 μM MG132. After processing 0, 1, and 3 h, samples were used for protein extraction and immunoblotting assay using anti-Strep (Cat # AF2924, Beyotime, China) and anti-Actin (Cat # A0840, Sigma, USA) antibodies.

## Subcellular localization

The coding sequence of *SlZHD8* was used to create 35S promoter-driven *SlZHD8-GFP* expression constructs. *35S::SlZHD8-GFP* was

transformed into *N. benthamiana* leaves via GV3101-mediated infiltration. The 4',6-diamidino-2-phenylindole (DAPI; Cat # C0065, Solarbio, China) was used as nuclear marker. Epidermal fluorescence was examined using Zeiss LSM 880 CLSM with the following parameters: GFP (excitation 488 nm, emission 510–550 nm) and DAPI (excitation 405 nm, emission 410–480 nm).

## Sugar transport function of SlSWEET12

The vectors pDR196-*SlSWEET12* and pDR196-*empty* (negative control) were transformed into the hexose transport mutant EBY4000 yeast strain (Lu et al, 2025). The transformed yeast cells were plated on minimal SC agar medium supplemented with 2% maltose as the sole carbon source. Positive clones were plated onto medium containing different carbon sources. A minor amount of isolated positive yeast colony was incubated within minimal SC liquid medium, enriched with 2% maltose at 30 °C for 16-18 h. When the stationary phase ($OD_{600} = 0.2–0.3$) was reached, the yeast cultures were transferred to fresh SC liquid medium and further incubated at 30 °C with agitation at 200 rpm for a period of 3–4 h until the $OD_{600} = 0.6$. At this stage, serial dilutions, ranging in factors of ($10^{-1}$, $10^{-2}$, $10^{-3}$, $10^{-4}$, $10^{-5}$) were prepared and subsequently dropped on minimal SC Agar containing 2% sugar (maltose, glucose, fructose) as the sole carbon source. The 2% maltose medium was used as a positive control.

## DNA-affinity purification sequencing (DAP-seq) assay

DAP-seq assay was performed as described previously (Sun et al, 2023) and using a DAP-seq Kit (Cat # D202009, Bluescape, China) according to the manufacturer's instructions. Fresh tomato seedlings were used for genomic DNA extraction. Fragmented gDNA was constructed into libraries using the NEXTFLEX Rapid DNA Seq Kit (PerkinElmer, USA). The coding sequence of *SlZHD8* was cloned into a pFN19K HaloTag T7 SP6 Flexi expression vector and was expressed using the TNT SP6 Coupled Wheat Germ Extract System (Promega, USA). Expressed proteins were directly captured using Magne Halo Tag Beads (Promega, USA). The SlZHD8-bound beads were incubated with adapter-ligated gDNA libraries. Eluted DNAs were sequenced on an Illumina Nova-Seq6000 with two technical duplicates. Beads without the addition of protein were taken as the input negative control DAP libraries. The fastp software default parameters were used to filter the raw data to obtain high-quality sequencing data/clean data for downstream analysis. Clean reads were mapped to the tomato reference genome Heinz 1706 version SL4.0 (The Tomato Genome Sequencing Consortium, 2012) to get unique mapped reads using BOWTIE2. MACS2 callpeak and IDR software were used to merge the peaks of the two biological duplicates with $Q < 0.05$. Motif discovery was performed using the MEME-ChIP software. The bound peaks were annotated using Homer software. The promoter regions were determined as the binding peaks within 2000 bp upstream of the transcription start sites. Finally, genes were aligned to tomato by blast with the criterion of identity over 90%. DAP-seq assay was conducted with two biological replicates.

## RNA sequencing (RNA-seq) assay

RNA-seq samples were extracted from fruit and root samples of WT and *slzhd8*, respectively. Every ten samples were mixed into one sample for a single biological replicate, and a total of three biological replicates were set up in RNA-seq assay. Isolated RNA was used for cDNA library construction with the NEBNext Ultra RNA Library Prep Kit (Illumina, USA) following the manufacturer's instructions. All libraries were subjected to paired-end 150-bp sequencing on the Illumina HiSeq 4000 platform. Illumina short paired-end reads produced previously were trimmed using Trimmomatic (v.0.36) with default parameters. The high-quality reads were aligned to the tomato reference genome Heinz 1706 version SL4.0 (The Tomato Genome Sequencing Consortium, 2012) using HISAT2 (v.2.0.4). The counts of reads aligned to each transcript were calculated with HTSeq (v.0.6.0) software, and fragments per kilobase of transcript per million mapped fragments were used to estimate the expression level of each gene.

## Variant calling and annotation

Resequencing data from 631 tomato accessions were obtained from previously published datasets (Zhou et al, 2022). Raw reads were subjected to quality control and adapter trimming using fastp (v1.0.1) (Chen, 2023) to remove adapters and low-quality bases. Clean reads were aligned to the Heinz 1706 genome (SL5.0) (Zhou et al, 2022) using bwa-mem2 (v2.3) with default parameters (https://github.com/bwa-mem2/bwa-mem2). PCR duplicate reads were marked using the MarkDuplicates tool in GATK (v4.6.2.0) (McKenna et al, 2010). SNPs and small indels were called per sample using GATK's HaplotypeCaller with default parameters. Joint genotyping across all samples was performed using GenotypeGVCFs with default parameters. Variants were filtered out using BCFtools (v1.22) based on the following criteria: QD < 3.0, MQ < 40.0, FS > 60.0, SOR > 3.0, MQRankSum_snp < -12.5, and ReadPosRankSum_snp < -8.0 (Danecek et al, 2021). Only bi-allelic SNPs were retained for subsequent analyses. Functional annotation of variants was conducted using SnpEff (v5.2t) with default parameters (Cingolani et al, 2012).

## Genome-wide detection of selective sweep regions

A total of 631 tomato accessions were divided into three groups: 78 *S. pimpinellifolium* L. (SP), 258 *S. lycopersicum* L. var. *cerasiforme* (SLC), 295 *S. lycopersicum* L. var. *lycopersicum* (SLL). We used the XP-CLR test to detect selective sweep regions with default parameters in 20-kb windows (Chen et al, 2010). To identify genes under selection, we ranked genomic regions based on their XP-CLR scores in descending order and designated the top 5% of regions as selective sweep regions. 1.68 and 1.93 were identified as the threshold of domestication and improvement sweeps, respectively. Genes overlapping these selective sweep regions were considered to be under selection.

## Statistical analyses

The statistical graphs were generated and processed by the GraphPad Prism 9 software. All values are presented as means ± SD, and the statistically significant differences between the control and experimental groups were determined by $t$ test, one-way ANOVA, or two-way ANOVA (Different letters indicate significant differences at $P < 0.05$; or $^{ns}P > 0.05$; $^*P < 0.05$, $^{**}P < 0.01$, $^{***}P < 0.001$, $^{****}P < 0.0001$).

## Data availability

Materials of Appendix Figs. S1–S17, Datasets EV1–EV10, and Source Data are available in the online version of this article. Raw data of DAP-seq and RNA-seq have been deposited to Sequence Read Archive (SRA; https://www.ncbi.nlm.nih.gov/sra) under the accession number PRJNA1306750.

The source data of this paper are collected in the following database record: biostudies:S-SCDT-10_1038-S44318-026-00708-0.

## Peer review information

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

## Acknowledgements

National Natural Science Foundation of China (32573008 and 32272697) and Taishan Scholars Program (tsqn202306139).

## Author contributions

**Jinghao Xu**: Data curation; Software; Formal analysis; Supervision; Methodology; Project administration. **Zhiliang Zhang**: Resources; Data curation; Software; Formal analysis; Supervision; Methodology; Project administration. **Jin-Wei Wei**: Conceptualization; Data curation; Software; Writing—review and editing. **Yingfang Zhu**: Resources. **Dan Zhao**: Data curation; Software; Methodology; Writing—review and editing. **Tianchen Xia**: Data curation. **Xiaoqian Liu**: Data curation; Software; Methodology. **Chengqiang Wang**: Software; Methodology. **Biao Gong**: Conceptualization; Supervision; Funding acquisition; Writing—original draft; Project administration; Writing—review and editing.

Source data underlying figure panels in this paper may have individual authorship assigned. Where available, figure panel/source data authorship is listed in the following database record: biostudies:S-SCDT-10_1038-S44318-026-00708-0.

## Disclosure and competing interests statement

The authors declare no competing interests.

