## [Peer Review File · The EMBO Journal]

Salt-stress-induced tomato sweetening involves an SISnRK2.6-SIZHD8 sugar accumulation cascade triggered by root-derived abscisic acid

Jinghao Xu, Zhiliang Zhang, Jin-Wei Wei, Yingfang Zhu, Dan Zhao, Tianchen Xia, Xiaoqian Liu, Chengqiang Wang, and Biao Gong

Corresponding author(s): Biao Gong (gongbiao@sdau.edu.cn)

Review Timeline:

Submission Date:	14th Aug 25
Editorial Decision:	8th Sep 25
Revision Received:	20th Oct 25
Editorial Decision:	21st Nov 25
Revision Received:	24th Nov 25
Accepted:	14th Jan 26

Editor: William Teale

Transaction Report:

Dear Prof. Gong,

Thank you again for the submission of your manuscript entitled "How salt stress sweetens tomatoes: root-derived ABA triggers a novel sugar accumulation cascade" and for your patience during the review process. We have now received reports from two referees, which I copy below.

As you can see from their comments, both thought that your manuscript could represent a useful and timely resource for the community. That said, all of them point out some issues that will require your attention before your manuscript can be published in The EMBO Journal.

Based on the overall interest expressed in the reports, however, I would like to invite you to address the comments of all referees in a revised version of the manuscript. I should add that it is The EMBO Journal policy to allow only a single major round of revision and that it is therefore important to resolve the main concerns at this stage. I believe the concerns of the referees are reasonable and addressable, but please contact me if you have any questions, need further input on the referee comments or if you anticipate any problems in addressing any of their points. I am available should you wish to discuss the reports and your response to them over Zoom. Please follow the instructions below when preparing your manuscript for resubmission.

I would also like to point out that as a matter of policy, competing manuscripts published during this period will not be taken into consideration in our assessment of the novelty presented by your study ("scooping" protection). We have extended this 'scooping protection policy' beyond the usual 3 month revision timeline to cover the period required for a full revision to address the essential experimental issues. Please contact me if you see a paper with related content published elsewhere to discuss the appropriate course of action.

Again, please contact me at any time during revision if you need any help or have further questions.

Thank you very much again for the opportunity to consider your work for publication. I look forward to your revision.

Best regards,

William

William Teale, Ph.D.
Editor
The EMBO Journal

When submitting your revised manuscript, please carefully review the instructions below and include the following items:

- 1) a .docx formatted version of the manuscript text (including legends for main figures, EV figures and tables). Please make sure that the changes are highlighted to be clearly visible.
- 2) individual production quality figure files as .eps, .tif, .jpg (one file per figure).
- 3) a .docx formatted letter INCLUDING the reviewers' reports and your detailed point-by-point response to their comments. As part of the EMBO Press transparent editorial process, the point-by-point response is part of the Review Process File (RPF), which will be published alongside your paper.
- 4) a complete author checklist, which you can download from our author guidelines ([https://wol-prod-cdn.literatumonline.com/pb-assets/embo-site/Author Checklist%20-%20EMBO%20J-1561436015657.xlsx](https://wol-prod-cdn.literatumonline.com/pb-assets/embo-site/Author%20Checklist%20-%20EMBO%20J-1561436015657.xlsx)). Please insert information in the checklist that is also reflected in the manuscript. The completed author checklist will also be part of the RPF.
- 5) Please note that all corresponding authors are required to supply an ORCID ID for their name upon submission of a revised manuscript.
- 6) We require a 'Data Availability' section after the Materials and Methods. Before submitting your revision, primary datasets produced in this study need to be deposited in an appropriate public database, and the accession numbers and database listed

under 'Data Availability'. Please remember to provide a reviewer password if the datasets are not yet public (see <https://www.embopress.org/page/journal/14602075/authorguide#datadeposition>). If no data deposition in external databases is needed for this paper, please then state in this section: This study includes no data deposited in external repositories. Note that the Data Availability Section is restricted to new primary data that are part of this study.

Note - All links should resolve to a page where the data can be accessed.

8) For data quantification: please specify the name of the statistical test used to generate error bars and P values, the number (n) of independent experiments (specify technical or biological replicates) underlying each data point and the test used to calculate p-values in each figure legend. The figure legends should contain a basic description of n, P and the test applied. Graphs must include a description of the bars and the error bars (s.d., s.e.m.).

9) We would also encourage you to include the source data for figure panels that show essential data. Numerical data can be provided as individual .xls or .csv files (including a tab describing the data). For 'blots' or microscopy, uncropped images should be submitted (using a zip archive or a single pdf per main figure if multiple images need to be supplied for one panel). Additional information on source data and instruction on how to label the files are available at .

10) We replaced Supplementary Information with Expanded View (EV) Figures and Tables that are collapsible/expandable online (see examples in <https://www.embopress.org/doi/10.15252/embj.201695874>). A maximum of 5 EV Figures can be typeset. EV Figures should be cited as 'Figure EV1, Figure EV2" etc. in the text and their respective legends should be included in the main text after the legends of regular figures.

12) Our journal encourages inclusion of *data citations in the reference list* to directly cite datasets that were re-used and obtained from public databases. Data citations in the article text are distinct from normal bibliographical citations and should directly link to the database records from which the data can be accessed. In the main text, data citations are formatted as follows: "Data ref: Smith et al, 2001" or "Data ref: NCBI Sequence Read Archive PRJNA342805, 2017". In the Reference list, data citations must be labeled with "[DATASET]". A data reference must provide the database name, accession number/identifiers and a resolvable link to the landing page from which the data can be accessed at the end of the reference. Further instructions are available at .

13) In order to increase the reproducibility and reach of your work, The EMBO Journal includes a table of reagents that were used in the study. Please provide this along with your revisions.

Further instructions for preparing your revised manuscript:

We realize that it is difficult to revise to a specific deadline. In the interest of protecting the conceptual advance provided by the work, we recommend a revision within 3 months (7th Dec 2025). Please discuss the revision progress ahead of this time with the editor if you require more time to complete the revisions. Use the link below to submit your revision:

Referee #1:

The manuscript by Xu et al identified the root-derived ABA as a key signal to enhance fruit sugar accumulation under salt stress, which could be transported over long distances from root to fruits. They also discovered the elevated fruit ABA will activate the kinase SnRK2.6, which phosphorylates the transcription factor ZHD8 to be degraded and weakens the binding activity of the downstream targets, SUS3 and SWEET12, eventually enhancing the sugar accumulation in fruits. Moreover, they found a beneficial ZHD8 haplotype, which is lost in the modern varieties, highlighting the potential application of ZHD8 in further tomato breeding. Overall, the authors carry out a number of detailed molecular and genetic studies to reveal the "stress enhances quality" mechanism

Major comments

1. In Fig 3g, the total amounts of SIZHD8-Strep in the five treatments were roughly the same. However, the total amount of phosphorylated SIZHD8-Strep combined with the unphosphorylated protein was much higher than that of the other treatments? Similar problems also exist in Fig 3i, 3j. These experiments require multiple repetitions, as required by many journals
2. In Fig 3K and I, in the protein degradation analysis assay, I noticed the protein level in the later time point (1h or 3h) was higher than that in the start (0h) point? Similarly, the cell-free degradation assay should be repeated at least 3 times, and the protein level should be quantitated.
3. In Fig 3m-p and Fig 5g, it's interesting to see the mutagenesis of ZHD8 lead to increased sugar content and salt tolerance without significantly affecting fruit weight, I wonder what's the other effect of the mutation on tomato, such as some yield-related traits, e.g. plant height, yield per plant, total fruit number? In addition, I noticed both SIZHD8T247A and SIZHD8T247D, which are considered to be no longer capable of being phosphorylated by SnRK2.6, could also respond to salt stress and increase the sugar content of the fruit in zhd8 mutant background, how can this be explained?
4. In Fig 6, the ZHD8HapB confers higher fruit sugar accumulation and stronger salt tolerance, however, the allele frequency of ZHD8HapB was gradually decreased during tomato domestication and improvement, I wondered whether ZHD8HapB has some negative impact on the yield or shelf-life, or does this locus have a linkage effect with other loci?
5. In the manuscript, all the salt stress analysis assays were conducted under moderate salt stress (0.1% NaCl), if the author could provide more field data, it would offer more practical assistance for tomato breeding.
6. What is the conservation of ZHD8 outside of tomato, across other fruit-bearing crops, or even more broadly across plants? I'm curious if the authors believe this will be a transferable strategy. If so, this dramatically increases the impact and implications of this study. Many other crops may potentially benefit.

Minor comments

1. In Fig 2g and I, the SSC and fructose content was also significantly increased in WT/not under salt stress, the description in L.203 "salt-induced fruit sugar accumulation was abolished in WT/not relative to WT/WT", is not accurate.
2. The model diagram of tomatoes in Fig 2a and Fig 7, is recommended to be replaced with an image that can show the actual growth of tomatoes.

Referee #2:

Salt stress-induced sugar accumulation is a critical determinant of both crop quality and stress tolerance. This manuscript elucidated a regulatory module involving SISnRK2.6-SIZHD8-SISWEET12/SISUS3, which integrates salt-induced ABA signaling with sugar metabolism. Specifically, the study proposed that tomato roots perceive salinity and activate ABA signaling, which is subsequently transported to the fruit. In the fruit, ABA signaling activates SISnRK2.6, leading to the modulation of SIZHD8 abundance. This, in turn, regulates the expression of SISWEET12 and SISUS3, thereby enhancing soluble sugar metabolism. By focusing on salt stress-mediated sugar accumulation in tomato fruit, the authors demonstrated that long-distance ABA transport promotes local fruit responses to salinity. These findings provide strong evidence for a mechanistic link between stress signaling and sugar metabolism and offer valuable insights for improving fruit quality in crops under salinity stress. I have the following suggestions to improve this discovery.

Major concerns:

1. Salt stress-induced ABA signaling in roots is generally triggered by osmotic stress and ion imbalance. At low salinity levels, plants typically grow normally, and ABA signaling is only weakly activated. In this manuscript, the authors selected 0.1% NaCl (approximately 18 mM), which appears too low to cause noticeable growth inhibition. The authors should provide further results on how different concentrations of salt treatments induce meaningful physiological or molecular responses.
2. In this manuscript, salt treatment induces the transport of ABA from roots to fruits and activates SnRK2.6. In Figure 2K, the authors demonstrated that ABA biosynthesis in roots is essential for SnRK2.6 activation, with exogenous ABA treatment in WT/WT plants serving as a positive control. However, an important question remains: in WT/not plants, can exogenous ABA application restore SnRK2.6 activity? Furthermore, does ABA treatment rescue sugar accumulation in the not mutant or in WT/not plants?
3. The authors verified the role of phosphorylation by complementing the *sizhd8* background with SIZHD8T247A and SIZHD8T247D. To further substantiate the function of SISnRK2.6, the authors should examine the phenotypes of non-phosphorylatable and phosphomimetic variants in the *sisnrk2.6* background.
4. SISWEET12 and SISUS3 are core components of sucrose metabolism in sink organs, including fruits and roots. The *sisweet12* and *sisus3* mutants exhibited heightened sensitivity to high-salinity stress at the seedling stage. Although SISWEET12 and SISUS3 both contribute to sugar accumulation, they function in different sink organs. This suggests that distinct strategies of sucrose metabolism may be employed to regulate salt-induced fruit maturation and salt tolerance. The authors should further discuss these organ-specific roles and their implications under salt stress.

Minor concerns:

1. The timing of salt treatment and its specific stage within the tomato growth cycle should be clearly described in the Materials and Methods section.
2. SIZHD8T247A represents the non-phosphorylatable form of SIZHD8, rather than a dephosphomimetic variant.
3. In the statistical analysis using two-way ANOVA, letters (a, b, c, d) should be used to visually annotate multiple comparisons, while appropriately reflecting the effects of different genetic materials.

Response to Editor and Reviewers

Dear Editor and Reviewers,

We are very pleased to receive your positive feedback and consider it an honor to have our work viewed so favorably. We have carefully reviewed all of the comments you provided and believe they are highly valuable for enhancing the quality of our manuscript.

Accordingly, in this revised version, we have fully addressed each of your suggestions and made corresponding improvements point by point. We sincerely appreciate your thorough and insightful guidance, which has greatly helped us refine this work.

In addition, we have also revised some grammatical, spelling errors, and inappropriate descriptions in this manuscript.

Thank you once again for your support and constructive input.

Sincerely,

Biao Gong

Point by point revision

Referee #1:

The manuscript by Xu et al identified the root-derived ABA as a key signal to enhance fruit sugar accumulation under salt stress, which could be transported over long distances from root to fruits. They also discovered the elevated fruit ABA will activate the kinase SnRK2.6, which phosphorylates the transcription factor ZHD8 to be degraded and weakens the binding activity of the downstream targets, SUS3 and SWEET12, eventually enhancing the sugar accumulation in fruits. Moreover, they found a beneficial ZHD8 haplotype, which is lost in the modern varieties, highlighting the potential application of ZHD8 in further tomato breeding. Overall, the authors carry

out a number of detailed molecular and genetic studies to reveal the "stress enhances quality" mechanism.

Major comments

1. In Fig 3g, the total amounts of SIZHD8-Strep in the five treatments were roughly the same. However, the total amount of phosphorylated SIZHD8-Strep combined with the unphosphorylated protein was much higher than that of the other treatments? Similar problems also exist in Fig 3i, 3j. These experiments require multiple repetitions, as required by many journals.

We sincerely appreciate your insightful comments on the need for more comprehensive data presentation. The initial submission format, which did not mandate the inclusion of all source data, resulted in our presentation of only a subset of the replicates. In response to your feedback, we have revised the manuscript to feature the highest-quality representative images alongside their statistical analysis in the main text. All supporting replicate data are now provided in the Supplementary Source Data section. Please see the revised content (Fig 3G,I,J) in the revised manuscript.

2. In Fig 3K and I, in the protein degradation analysis assay, I noticed the protein level in the later time point (1h or 3h) was higher than that in the start (0h) point? Similarly, the cell-free degradation assay should be repeated at least 3 times, and the protein level should be quantitated.

During the experimental design and execution, we included three replicates. As per the initial submission guidelines, which did not require the submission of all source data, only a subset was originally presented. In response to your comments, we have now selected higher-quality, representative images for the main text, along with their relevant statistical results. The remaining replicate data have been included in the Supplementary Source Data section. Please refer to the updated Fig. 3K,L in the revised manuscript.

3. In Fig 3m-p and Fig 5g, it's interesting to see the mutagenesis of ZHD8 lead to increased sugar content and salt tolerance without significantly affecting fruit weight, I wonder what's the other effect of the mutation on tomato, such as some yield-related traits, e.g. plant height, yield per plant, total fruit number? In addition, I noticed both SIZHD8T247A and SIZHD8T247D, which are considered to be no longer capable of being phosphorylated by SnRK2.6, could also respond to salt stress and increase the sugar content of the fruit in *zhd8* mutant background, how can this be explained?

Regarding the agronomic traits: Compared with the WT, the *slzhd8* mutant has only a minor impact on key agronomic characteristics of tomato, such as growth phenotype, biomass, yield, photosynthesis, chlorophyll content, mineral nutrient absorption capacity, reactive oxygen species scavenging capacity. Although we conducted related investigations—including assessments of

agronomic traits and physiological metabolism—we decided during the writing process that these data were not directly relevant to the core focus of this study. Therefore, they were excluded from the previous manuscript. In the revised version, we have instead incorporated these data into the discussion, where they serve as supplementary information supporting the broader context of our work. For details, please refer to the Supplementary Fig. S15 and the discussion (Lines 537-544) in the revised manuscript.

Regarding the effect of the T247 mutation on sugar accumulation: In the comparison between Salt and Control treatments in the WT background, the difference was highly significant ($****P < 0.0001$). In contrast, in the *slzhd8/SIZHD8^{T247A}* and *slzhd8/SIZHD8^{T247D}* background, the difference between Salt and Control treatments was only marginally significant ($*P < 0.05$), indicating that the induction of fruit sugar accumulation by salt treatment was markedly attenuated. It is worth noting, however, that under both T247A and T247D mutations, salt stress still slightly promoted sugar accumulation—a pattern consistent with that observed in the *slsnrk2.6* mutant (Fig. 3M-P). This suggests that while the *SISnRK2.6-SIZHD8* signaling pathway plays a major role in regulating sugar accumulation under salt stress, it is not the sole contributing mechanism; other factors are also involved. Additionally, in response to “Minor concern (3)” raised by Reviewer #2, we have now marked the treatment–genotype interaction patterns in the ANOVA using different letters. Under this new analytical approach, the effect of salt treatment on fruit sugar accumulation in the *T247A* and *T247D* mutants was not statistically significant ($P > 0.05$). Overall, different modes of analysis consistently demonstrate that mutation at *T247* weakens the induction of fruit sugar accumulation by salt treatment.

4. In Fig 6, the ZHD8HapB confers higher fruit sugar accumulation and stronger salt tolerance, however, the allele frequency of ZHD8HapB was gradually decreased during tomato domestication and improvement, I wondered whether ZHD8HapB has some negative impact on the yield or shelf-life, or does this locus have a linkage effect with other loci?

Based on our field observations, neither the *SIZHD8* mutation nor its different haplotypes had any significant effect on fruit yield or the ripening process (Please see Supplementary Fig. S15). Consistent with these phenotypic results, no related genes (regarding growth and yield) were identified in the transcriptome data of WT and *slzhd8* mutant.

Following your suggestion, we conducted a genome-wide analysis of linkage disequilibrium (LD) decay in the tomato genetic population. The LD decay pattern showed the strongest extent in *SLL*, an intermediate level in *SLC*, and the weakest in *SP*, reflecting the cumulative effects of breeding history. The high LD observed in *SLL* is consistent with historical directional selection and

genetic drift during domestication (Supplementary Fig. S17A). Notably, ZHD8 lay outside the contiguous association block shared by neighboring loci, suggesting that it may function as an independent genetic determinant of the trait (Supplementary Fig. S17B). These insights have been incorporated into the Discussion section (Lines 581-588) of the revised manuscript.

5. In the manuscript, all the salt stress analysis assays were conducted under moderate salt stress (0.1% NaCl), if the author could provide more field data, it would offer more practical assistance for tomato breeding.

Regarding the issue of salt concentration, which was also raised by Reviewer #2 under "Major concerns (1)", we performed additional experiments to evaluate key physiological and biochemical parameters—including ABA content, *SISnRK2.6* kinase activity, *SIZHD8* phosphorylation status, *SIZHD8* protein stability, and transcript levels of *SISUS3* and *SIWEET12*—across the original salt treatment groups (Control, 0.1% NaCl, 0.2% NaCl, 0.3% NaCl) presented in Supplementary Fig. S1. These results support the generality of the *SISnRK2.6–SIZHD8–SISUS3/SIWEET12* pathway under varying salt stress conditions. In addition, we have provided further data on field agronomic traits associated with *SIZHD8* and its haplotypes, as detailed in Supplementary Fig. S15.

6. What is the conservation of ZHD8 outside of tomato, across other fruit-bearing crops, or even more broadly across plants? I'm curious if the authors believe this will be a transferable strategy. If so, this dramatically increases the impact and implications of this study. Many other crops may potentially benefit.

This is an insightful suggestion. We identified ZHD8 homologous proteins across all 16 fruit-bearing crops examined (see figure below), indicating that this transcription factor is relatively conserved. However, the T247 site is only conserved in tomato and potato, suggesting species-specific variation in the phosphorylation pathway mediated by this residue. Interestingly, we observed that while the T247 is not conserved in other plants, the adjacent T248 is highly conserved. Given the proximity of these two residues and the overall conservation of this region in ZHD8, it is plausible that T248 may serve as a potential phosphorylation site for SnRK2.6 or other SnRK family members in other crops. These observations, however, remain speculative and extend beyond the scope of the current study. Therefore, we have chosen to communicate this finding only in "Response to Reviewers" and will not include it in the revised manuscript.

```

Solanum lycopersicum (Tomato)      .....MELNINNTTAAIT.....TVKTPELAETETP.....SRI..QQPKPFFSNGVLKRKNHHHP.....V.....VVVYKECLKN    62
Solanum tuberosum (Potato)         .....MELNTNT.TTAPAAA.....AIKTPELAETETP.....SRI..QQPKPFFSNGVLKRKN.HHP.....V.....VVVYKECLKN    60
Solanum melongena (Eggplant)      .....MDLSTNA.TTTAAAAATSSAIKTPELAETETP.....SRI..QQPKPFFSNGVFKHKH.HHP.....V.....VVIYRECLKN    65
Capsicum annuum (Chili)           .....MDLTKTT.TTTT.....TTTSVKTPE.AETETP.....TRI..QQQKPPFFSDGVLKRKS.HHQ.....V.....HRHLAVTYRECLKN    64
Malus domestica (Apple)           .....MDITPSITTTNNNT.....ASTKSPEADESETP.....TRIQ.QPLKPLSFS.NGVLKRHNP.T.....HHLHHQNIPTIPV.....V.....VITYKECLKN    73
Fragaria × ananassa (Fragaria)    .....MDIAPSIITTTA.....STKSPEADESPTP.....TRI.QPAKPISFS.NGVLKRHNPTHHHHSHHHSNIPTVTV.....V.....VSIYRECLKN    73
Prunus persica (Peach)            .....MDITPSITTTNNNT.....TSTKSPEADESETP.....TRI.QPAKPLSFS.NGVLKRHNP.HPHHHHHHHHNIPTIPV.....V.....IVTYKECLKN    75
Actinidia chinensis (Kiwi fruit)  .....MDISTATVY.....ACVKTPPEASEAP.....TRIQVPGKPLCFT.NGVLKRHHP.....T.....HHVNSPPA.....V.....VITYKECLKN    63
Cerasus pseudocerasus (Cherry)    .....MDITPSITTTNNNT.....TSTKSPEADESETP.....TRI.QPAKPLSFS.NGVLKRHX.....T.....PV.....IVTYKECLKN    59
Citrus sinensis (Sweet orange)    .....MDITPTNNLTLNTN.....TNSKSPETDITDNHQGTATRI..HSAKPLTFT.NGVLKRHP.QQHQRHRRHHHHHHHP.....V.....VITYKECLKN    80
Cocos nucifera (Coconut)         .....MDPATAK.....ILD.THTKPKPFFSPNGSLKR..HHRGLSSPVEA.....AGAA.....DFLYRECLKN    51
Cucumis sativus (Cucumber)       .....APITTSN.....NTKSPDPDSPTP.....TKIP.P.....SFT.NGVLKRHHH.....HHHHRPSSVT.....V.....VSIYKECLKN    60
Vitis vinifera (Grape)           MEVSAAAATAVADTGGAAVAVG.....GGVKSPAEATEPTP.....TOI..QPRKGLSLT.NGVLKRHQHHHHHHHFAAPQ.....V.....VVIYKECLKN    79
Musa acuminata (Banana)          MDVSGREEEMPTMASACV.....GDHGHNSSIHDH.....PPI.HHSTNGPPPLSTIV.TTEDHHH..HG.....KKGV.....VVKYRECLKN    71
Phoenix dactylifera (Phoenix)     MDLSGHEGEIPIITSAYV.....GGHGTIIHDTP.....PLH..HHPSNGPPPPPLPTATSEDOHQ..HPTNPSYSSKGV.....V.....VLKYRECLKN    79
Punica granatum (Pomegranate)    .....MEITSTAA.....HASKSPEPEIDTP.....TRI..QPAKPVPTCTNGVLKR.....V.....VITYRECLKN    61
Consensus                        y ec kn

```

```

Solanum lycopersicum (Tomato)      HAAALGTHAVDGCGEFPIPAANPADPDSLKCAACGCHRNFRHR.....EPEEPPPIA.....TAAIEYQPH.HRHH.PPPPR..G.    134
Solanum tuberosum (Potato)         HAAALGTHAVDGCGEFPIPAANPADPDSLKCAACGCHRNFRHR.....DPEEPPPIA.....TAAIEYQPH.HRHH.PPPPC..G.    132
Solanum melongena (Eggplant)      HAAALGTHAVDGCGEFPIPAANPADPDSLKCAACGCHRNFRHR.....DPEEPPPIA.....TAAIEYQPH.HRHH.PPPPR..G.    137
Capsicum annuum (Chili)           HAAALGHAVDGCGEFPIPSANPTDPSLKCAACGCHRNFRHR.....EPEEVPVLPAPPIS.....TAAIEYQPH.HRHH.PPPPPPCG.    144
Malus domestica (Apple)           HAAALGHAVDGCGEFPSPAANLADPDSLKCAACGCHRNFRHR.....DPEDPVQNPATA.....THVIEYQPH.HRHH.PPPPT..HP    151
Fragaria × ananassa (Fragaria)    HAAALGHAVDGCGEFPSPSTPSDPSLKCAACGCHRNFRHR.....DPEDNIPAV..APA.....THVIEYQPH.HRHH.PPPPI..I.    148
Prunus persica (Peach)            HAATLGHALDGCGEFPSPPTAIPDPSLKCAACGCHRNFRHR.....DPEDPMPSSAAAT.....THVIEYQPH.HRHH.PPPPT..HG    153
Actinidia chinensis (Kiwi fruit)  HAAALGHAVDGCGEFPSPAANLADPDSLKCAACGCHRNFRHR.....DPEEPPPIA.....QNLIEYQPH.HRHH.PPPPR..RV    139
Cerasus pseudocerasus (Cherry)    HAATLGHALDGCGEFPSPPTAIPDPSLKCAACGCHRNFRHR.....DPEEPPPIA.....QNLIEYQPH.HRHH.PPPPT..RG    137
Citrus sinensis (Sweet orange)    HAAALGHAVDGCGEFPSPPTAIPDPSLKCAACGCHRNFRHR.....DPEEPPPIA.....TATIEYQPH.HRHH.PPPPV..TG    160
Cocos nucifera (Coconut)         HAAALGHAVDGCGEFPSPAANPADPDSLKCAACGCHRNFRHR.....PEFLFHHSDNGTGGGEGEDDMMAGHDEHDD.GRDDEE.EGSA.DGRR.....    142
Cucumis sativus (Cucumber)       HAATLGHALDGCGEFPSPSPTSDPSLRCACGCHRNFRHR.....DPEEPPPIA.....THVIEYQPH.HRHH.PPPPL..A.    139
Vitis vinifera (Grape)           HAAALGHAVDGCGEFPSPAANLADPDSLKCAACGCHRNFRHR.....EPDPPP.....PT.....THVIEYQPH.HRHH.PPPPR..LV    150
Musa acuminata (Banana)          HAASIGSATDGCGEFPGSEEGTPEA..LKCAACGCHRNFRHR.....EKEGEP.SC..DCF.....HPFRGRKVMVGQK.FLVSG..S.    144
Phoenix dactylifera (Phoenix)     HAASIGSATDGCGEFPGSEEGTLEA..LKCAACGCHRNFRHR.....EVEGESSC..DCF.....HHLKRKVL.GQKG.LLISG..P.    152
Punica granatum (Pomegranate)    HAAALGHAVDGCGEFPSPAANLADPDSLKCAACGCHRNFRHR.....DPEEPPPIA.....HA..TEYHPH.HRHH.PPPR..    135
Consensus                        haa g a gcgef p l c ac chrnfr

```

```

Solanum lycopersicum (Tomato)      DH..GSPN.SPSPPISSAYYPASAPHMLLALSSG.....FSGEK..NQNLPTSTTPMAVANS.NGRKFRFTKFTPDQKIKMLEFAE    210
Solanum tuberosum (Potato)         DH..SSPN.SPSPPISSAYYPASAPHMLLALSSG.....FSGEK..NQNLPTSTTPMAVANS.NGRKFRFTKFTPDQKIKMLEFAE    208
Solanum melongena (Eggplant)      DH..SSPN.SPSPPISSAYYPASAPHMLLALSSG.....FSGEK..NQNLPTSTTPMAVANS.NGRKFRFTKFTPDQKIKMLEFAE    216
Capsicum annuum (Chili)           DH..SSPN.SPSPPISSAYYPASAPHMLLALSSG.....FSGEK..NQNLPTSTTPMAVANS.NGRKFRFTKFTPDQKIKMLEFAE    218
Malus domestica (Apple)           GN..RSPS.SASPPPISSAYYP.SAPHMLLALSSG.....HENALAGA..N..NNAVAVQMPVMSRPNARKRFRFTKFTDQKIKMLEFAE    229
Fragaria × ananassa (Fragaria)    GH..RSPN.SASPPPISSAYYP.SAPHMLLALSSG.....LSDN..P..NNHGGQIVS.PGPNRKRFRFTKFTDQKIKMLEFAE    221
Prunus persica (Peach)            GN..RSPN.SASPPPISSAYYP.SAPHMLLALSSG.....HENALGGP..N..NNHSPQPIV.SPSNARKRFRFTKFTDQKIKMLEFAE    230
Actinidia chinensis (Kiwi fruit)  GH..SSPN.SASPPPISSAYYP.SAPHMLLALSSG.....LAAAPPE..NNPSTIPP..NNRKRFRFTKFTDQKIKMLEFAE    211
Cerasus pseudocerasus (Cherry)    GN..RSPN.SASPPPISSAYYP.SAPHMLLALSSG.....HENALGGP..N..NNHSPQPIV.SPSNARKRFRFTKFTDQKIKMLEFAE    214
Citrus sinensis (Sweet orange)    PPSRSPS.SASPPPISSAYYP.SAPHMLLALSSG.....HENALGGP..N..NNHSPQPIV.SPSNARKRFRFTKFTDQKIKMLEFAE    253
Cocos nucifera (Coconut)         ...RGRS.SASPPPISSAYYP.SAPHMLLALSSG.LPA.....PSLAR..PVAVSTVAVLP.....RKRFRFTKFTDQKIKMLEFAE    208
Cucumis sativus (Cucumber)       GN..RSPN.SASPPPISSAYYP.SAPHMLLALSSG.....LSDN..P..NNHGGQIVS.PGPNRKRFRFTKFTDQKIKMLEFAE    214
Vitis vinifera (Grape)           RP..RSPN.SPSPPISSAYYP.SAPHMLLALSSG.....ISGP..P..ENAPPISSP.ASANGRKRFRFTKFTDQKIKMLEFAE    223
Musa acuminata (Banana)          DA..FGY.PAGNSLPRVW.....MPLGAMQTESDEMEGV.GGMVMP.....AMVYKFRFTKFTDQKIKMLEFAE    209
Phoenix dactylifera (Phoenix)     EA..FGY.PAGNSLPRVW.....MPLGAMQTESDEMEGV.GGMVMP.....AMVYKFRFTKFTDQKIKMLEFAE    223
Punica granatum (Pomegranate)    ...RSPS.SASPPPISSAYYP.SAPHMLLALSSG.....LSDN..P..NNHGGQIVS.PGPNRKRFRFTKFTDQKIKMLEFAE    207
Consensus                        krrft f qk m e

```

```

Solanum lycopersicum (Tomato)      KVEIKYQKRDDELVNFCEIEVEKGVLKVMHNNKNTSISGKGLD....QPNTDGNHNQNGNSNYVNGFCIVDRNNTT..HHHD.NTDESEH...    298
Solanum tuberosum (Potato)         KVEIKYQKRDDELVNFCEIEVEKGVLKVMHNNKNTSISGKGLD....QPITGNQD.HQNGTNNINLVNFCIVSRNNTT..HPDNNTSEFH...    296
Solanum melongena (Eggplant)      KVEIKYQKRDDELVNFCEIEVEKGVLKVMHNNKNTSISGKGLD....QSSTGHQV.GNTNVV..VNGFCIVTKNKT..HD..NTDESEH...    298
Capsicum annuum (Chili)           RVEIKYQKRDDELVNFCEIEVEKGVLKVMHNNKNTSISGKGLD....Q....HSG.NIANNI..VNGFCIASRKDGH..YD..KPDSEFH...    294
Malus domestica (Apple)           RVGKYQKRDDEIVREFCNEIEVEKGVLKVMHNNKNTFSKRDVNL...GGAGGRAGLSR.....PSF.....LLESHHHHNG    301
Fragaria × ananassa (Fragaria)    RVGKYQKRDDEIVREFCNEIEVEKGVLKVMHNNKNTFSKRDVNL...SGSGGALNGISA.....TVT...ARIP..NNTS.LDEQLH...NN    298
Prunus persica (Peach)            RVGKYQKRDDEIVREFCNEIEVEKGVLKVMHNNKNTFSKRDVNL...GSGGGLSAGVSR.....PNI.....LLEEAA..NN    299
Actinidia chinensis (Kiwi fruit)  RLEIKYQKRDDEIVREFCNEIEVEKGVLKVMHNNKNTFSKRDVNL...ENGINANLEN.....TPT.....HHHHHHH.DQNFNHSVSA    290
Cerasus pseudocerasus (Cherry)    RVGKYQKRDDEIVREFCNEIEVEKGVLKVMHNNKNTFSKRDVNL...GSGGGLSAGVSR.....PNI.....LLEEAA..NN    299
Citrus sinensis (Sweet orange)    RLVEIKYQKRDDELVRDFCNEIEVEKGVLKVMHNNKNTFSKRDVNL...AGSGSAGSAGG.....IGR.....INLDDD..NT    372
Cocos nucifera (Coconut)         RLVGKYQKRDDELVRDFCNEIEVEKGVLKVMHNNKNTFSKRDVNL...GSGGGLSAGVSR.....PNI.....LLEEAA..NN    329
Cucumis sativus (Cucumber)       RVGKYQKRDDELVAEFCNEIEVEKGVLKVMHNNKNTFSKRDVNL...SRTSLDDNENNENENTENTET..NETN..ENND.NTPHTM...ET    308
Vitis vinifera (Grape)           KVGKRLQKEESVQQFCQETGKRRVLRVLMVHNNKNTLQKIS.LQLE.....V.....    257
Musa acuminata (Banana)          KVGKRLQKEESVQQFCQETGKRRVLRVLMVHNNKNTLQKIS.LQLE.....V.....    272
Phoenix dactylifera (Phoenix)    QVGKIKYQKRDDELVNFCEIEVEKGVLKVMHNNKNTSISGKGLD....QPNTDGNHNQNGNSNYVNGFCIVDRNNTT..HHHD.NTDESEH...    298
Punica granatum (Pomegranate)    QVGKIKYQKRDDELVNFCEIEVEKGVLKVMHNNKNTSISGKGLD....QPNTDGNHNQNGNSNYVNGFCIVDRNNTT..HHHD.NTDESEH...    281
Consensus                        w q c v v kvwmhnnk

```

```

Solanum lycopersicum (Tomato)      IHHE.S.SMNDNKKENSSFGANNVVNTNGSSSSS.....    332
Solanum tuberosum (Potato)         IHHE.S.SMNDNKKENSSFGANNVVNTNGSSSSS.....    329
Solanum melongena (Eggplant)      LHHE.S.SMNDNKKENSSFGANNVVNTNGSSSSS.....    330
Capsicum annuum (Chili)           LLRES.S.SMNDNKKENSSFGANNVVNTNGSSSSS.....    324
Malus domestica (Apple)           TNGNTNGNNDDEEDDEEDD.....DQND...NKNVGNPNNHVYQGADGGG.....NGSSSSS.....    353
Fragaria × ananassa (Fragaria)    MNGNN.NMNTNTDDEEDDHPHHHHN.....GGSLGNIPNPNHHYQVGRDHPHGTNGSSSSS    359
Prunus persica (Peach)            GNGNG.NGNGNNDDEEDD.....HNDH...NNSDMQLNHYQGTGGAGHVTNGSSSSS.....    354
Actinidia chinensis (Kiwi fruit)  HVVATNGSSSSS.....    302
Cerasus pseudocerasus (Cherry)    GNGNG.NGNGNNDDEEDD.....QNDNNTNNSDMQLNHYQGTGGAGHVTNGSSSSS.....    337
Citrus sinensis (Sweet orange)    GNDNI.NNSKSGDDQDQDEEENNNNNGVRLNHQFGSATESAAHVANANG.....GSSSSS.....    379
Cocos nucifera (Coconut)         ...NGSSSSS.....    308
Cucumis sativus (Cucumber)       IHHTP.NHNHTNPNPNNDTTTAAHLATNGSSSSS.....    292
Vitis vinifera (Grape)           IHHTP.NHNHTNPNPNNDTTTAAHLATNGSSSSS.....    341
Musa acuminata (Banana)          .....    257
Phoenix dactylifera (Phoenix)    .....    272
Punica granatum (Pomegranate)    TNGSSSSS.....    289
Consensus

```

Minor comments

1. In Fig 2g and I, the SSC and fructose content was also significantly increased in WT/not under salt stress, the description in L.203 "salt-induced fruit sugar accumulation was abolished in WT/not relative to WT/WT", is not accurate.

We have revised this inappropriate description.

2. The model diagram of tomatoes in Fig 2a and Fig 7, is recommended to be replaced with an image that can show the actual growth of tomatoes.

The schematic description in the original Fig. 2A is very simple. Even without referring to that panel, the methodology we employed remains clearly understandable. In the revised version, we have removed the original model description to improve visual clarity and layout compactness.

As for the tomato illustrations in the model diagram of Fig. 7, we experimented with replacing them with actual plant images in the style of Supplementary Fig. S15A. However, this made the figure appear visually cluttered and less aesthetically cohesive. Therefore, we prefer to retain the simplified cartoon-style tomatoes, which are consistent with the original submission and are widely used in many botanical publications. This approach helps maintain a clean and professional visual presentation.

Referee #2:

Salt stress-induced sugar accumulation is a critical determinant of both crop quality and stress tolerance. This manuscript elucidated a regulatory module involving SISnRK2.6-SIZHD8-SISWEET12/SISUS3, which integrates salt-induced ABA signaling with sugar metabolism. Specifically, the study proposed that tomato roots perceive salinity and activate ABA signaling, which is subsequently transported to the fruit. In the fruit, ABA signaling activates SISnRK2.6, leading to the modulation of SIZHD8 abundance. This, in turn, regulates the expression of SISWEET12 and SISUS3, thereby enhancing soluble sugar metabolism. By focusing on salt stress-mediated sugar accumulation in tomato fruit, the authors demonstrated that long-distance ABA transport promotes local fruit responses to salinity. These findings provide strong evidence for a mechanistic link between stress signaling and sugar metabolism and offer valuable insights for improving fruit quality in crops under salinity stress. I have the following suggestions to improve this discovery.

Major concerns:

1. Salt stress-induced ABA signaling in roots is generally triggered by osmotic stress and ion imbalance. At low salinity levels, plants typically grow normally, and ABA signaling is only weakly activated. In this manuscript, the authors selected 0.1% NaCl (approximately 18 mM), which appears too low to cause noticeable growth inhibition. The authors should provide further results on how different concentrations of salt treatments induce meaningful physiological or molecular responses.

Before addressing this question, please allow me to provide some context regarding the use of moderately saline soil or growth media to improve tomato quality. The concept that “stress enhances quality” is both a long-standing observation in agricultural practice and a meaningful biological phenomenon. However, the stress applied is usually mild to moderate. Excessively strong stress not only inhibits growth and reduces yield—thereby diminishing economic value—but may also impair fruit quality. Accordingly, this study represents a shift in focus, addressing the practical demands of agricultural production for yield and growth, rather than confining itself to the theoretical scope of salt stress.

In terms of soil salinity, a salt content of 0.1-0.3% NaCl is classified as mildly saline, under which some glycophytic species such as tomato can still complete their life cycle. A salinity level of 0.3-0.6% NaCl is considered moderately saline, and most glycophytes cannot survive under such conditions; beyond 0.6% NaCl, the environment is severely saline, permitting only a limited number of halophytic plants to grow. Therefore, tomato cultivation in saline soils is typically conducted in mildly saline conditions, with salt concentrations below 0.2% being the most common.

It is worth noting that as early as the 1990s, Japan pioneered the use of diluted seawater in soilless cultivation systems to increase mineral content, leading to the production of high-quality tomatoes and coining the term “fruit tomato”. This cultivation model remains in use today and has been further refined—for instance, by replacing NaCl with nutrient salts like KNO₃, which not only raises electrical conductivity but also supplies essential minerals. Whether through mild-salinity irrigation or mineral supplementation, these practices are implemented with the goal of minimizing impacts on growth and yield, while their beneficial effect on fruit quality has been consistently demonstrated in production.

Returning to the point you raised: we cannot assume that just because 0.1% NaCl has minimal impact on growth, it equally minimally affects metabolic processes. Metabolic responses are often more sensitive than phenotypic traits, a notion supported by numerous metabolomic studies. That said, we fully agree that as part of rigorous scientific research, it is necessary to substantiate key physiological and biochemical changes across different NaCl treatments.

Accordingly, under the same salt conditions previously shown in Supplementary Fig. S1 (Control, 0.1% NaCl, 0.2% NaCl, 0.3% NaCl), we have now measured ABA content, *SISnRK2.6* kinase activity, *SIZHD8* phosphorylation status, *SIZHD8* protein stability, and transcript levels of *SISUS3* and *SIWEET12*—core components closely related to this study. Together with previously assessed traits such as plant growth and fruit sugar content, these new data strongly support the conserved role of the *SISnRK2.6-SIZHD8-SISWEET12/SISUS3* pathway under varying salt stress

conditions. These data have been incorporated into the discussion section (Lines 516-523) of the revised manuscript.

2. In this manuscript, salt treatment induces the transport of ABA from roots to fruits and activates SnRK2.6. In Figure 2K, the authors demonstrated that ABA biosynthesis in roots is essential for SnRK2.6 activation, with exogenous ABA treatment in WT/WT plants serving as a positive control. However, an important question remains: in WT/not plants, can exogenous ABA application restore SnRK2.6 activity? Furthermore, does ABA treatment rescue sugar accumulation in the not mutant or in WT/not plants?

Based on your valuable suggestions, we have conducted experiments to measure sugar content and SnRK2.6 kinase activity in both WT/WT and WT/*not* grafting plants under Control, Salt, and Salt + ABA treatment conditions. The corresponding results are now provided in Fig. 2C-F of the revised manuscript.

3. The authors verified the role of phosphorylation by complementing the *sizhd8* background with SIZHD8T247A and SIZHDT247D. To further substantiate the function of SISnRK2.6, the authors should examine the phenotypes of non-phosphorylatable and phosphomimetic variants in the *sisnrk2.6* background.

Based on your valuable suggestions, we introduced the wild-type *SIZHD8* gene, as well as its *SIZHD8*^{T247A} and *SIZHD8*^{T247D} mutant versions using PVX approach, into both the *slzhd8* mutant and *sisnrk2.6/slzhd8* double mutant backgrounds. We then measured and compared the fruit sugar content of these different genotypes under both control and salt stress conditions. The corresponding results are now provided in Supplementary Fig. S9 of the revised manuscript.

4. SISWEET12 and SISUS3 are core components of sucrose metabolism in sink organs, including fruits and roots. The *sisweet12* and *sisus3* mutants exhibited heightened sensitivity to high-salinity stress at the seedling stage. Although SISWEET12 and SISUS3 both contribute to sugar accumulation, they function in different sink organs. This suggests that distinct strategies of sucrose metabolism may be employed to regulate salt-induced fruit maturation and salt tolerance. The authors should further discuss these organ-specific roles and their implications under salt stress.

In response to your insightful comments, we performed tissue-specific expression analysis of the core signaling pathway components, *SISnRK2.6*, *SIZHD8*, *SISUS3*, and *SISWEET12* (Supplementary Fig. S16). The resulting expression patterns aligned well with the central premise of our study. Consequently, we have integrated this gene expression data with the relevant aspects of sugar partitioning into the Discussion section (Lines 552-561) of the revised manuscript.

Minor concerns:

1. The timing of salt treatment and its specific stage within the tomato growth cycle should be clearly described in the Materials and Methods section.

We have revised the relevant descriptions to make the processing method clearer.

2. SIZHD8T247A represents the non-phosphorylatable form of SIZHD8, rather than a dephosphomimetic variant.

We have corrected this mistake.

3. In the statistical analysis using two-way ANOVA, letters (a, b, c, d) should be used to visually annotate multiple comparisons, while appropriately reflecting the effects of different genetic materials.

The ANOVA has been updated as suggested.

Dear Prof. Gong,

We have now received re-review reports from two referees, which I have included below. As you will see, you have addressed many of their concerns satisfactorily; however, I would like you, in response to Referee #2, to (in the main text) acknowledge and discuss the limitations of your work in the context of sample size and the potential for accumulation of NaCl in the soil (in the absence of these measurements being carried out). Please also consider the minor revisions recommended by Reviewer #1. Before I can finally accept the manuscript, there are also some remaining editorial points which need to be addressed. In this regard would you please:

- remove figures from the main manuscript and upload them as individual, high-resolution figure files; ensure track changes is not used,
- state employment by and/or association with a biotech company in the disclosure and competing interests statement,
- acknowledge funding by the National Natural Science Foundation of China (32573008) and Taishan Scholars Program (tsqn202306139) both in our online submission system and the manuscript's "Acknowledgements" section,
- reduce the number of keywords to five,
- use the first ten names + et al. in the reference section for longer author lists,
- remove the AC/CrediT section from the manuscript text,
- include a callout for Figure S14 (as Appendix Figure S14) in the main text,
- complete the middle column (D) in our author checklist,
- update source file names, titles, legends and manuscript callouts to Dataset EV1-EV10 instead of Dataset S1-S10; they should be uploaded individually as Dataset files with legends in a separate tab/sheet in each Excel file,
- convert the Appendix file to a PDF format; Supplementary Figures should be compiled in Appendix PDF - the title page should contain "Appendix for 'How salt stress sweetens tomatoes: root-derived ABA triggers a novel sugar accumulation cascade'" and a table of contents included with the page numbers for the listed items; nomenclature should be Appendix Figure Sx and Appendix Table Sx throughout the manuscript and Appendix PDF,
- include a Reagents and Tools table,
- ensure dataset PRJNA1306750 is publicly available,
- provide a URL for PRJNA1306750 in the data availability statement,
- provide exact p values are not provided in the legends of figures 1B, C, D, E, F, G, I, J, K, L; 2A, B, C, D, E, F, G, H, I, J; 3 H-L, M-P; 4F, G, K, L, M; 5A-C, D-F, H-K; 6D, G, H, I, J, M, N,
- specify the statistical test that was used for data analysis in the legend of figure 4D,
- define box plots in terms of minima, maxima, centre, bounds of box and whiskers, and percentile in the legends of figures 1B, C, I, J, K, L; 2C, D, E, G, H, I, J; 3M-P; 5A-F; 6G-J, and
- correct the section order as follows: Title page - Abstract - Keywords - Introduction - Results - Discussion - Methods - Data Availability - Acknowledgements - Disclosure and Competing Interests Statement - References - Figure Legends - Table(s) - Expanded View Figure Legends.

We include a synopsis of the paper (see <http://emboj.embopress.org/>). Please provide me with a general summary image, a two sentence statement and 3-5 bullet points that capture the key findings of the paper.

I am looking forward to receiving your revised manuscript.

EMBO Press is an editorially independent publishing platform for the development of EMBO scientific publications.

Best wishes,

William Teale

William Teale, PhD
Editor
The EMBO Journal
w.teale@embojournal.org

Read our guidance for manuscript revisions and related editorial policies: <https://link.springer.com/journal/44318/submission-guidelines#cms-Revised-submissions>

<https://media.springernature.com/original/springer-cms/rest/v1/content/27825798/data/v1>

- a point-by-point response to the referees' comments, with a detailed description of the changes made (as a word file).
- a word file of the manuscript text.
- individual production quality figure files (one file per figure)
- a complete author checklist
- Expanded View files (replacing Supplementary Information)
- a Reagents and Tools Table as part of the Methods section

Please remember: Digital image enhancement is acceptable practice, as long as it accurately represents the original data and conforms to community standards. If a figure has been subjected to significant electronic manipulation, this must be noted in the figure legend or in the 'Methods' section. The editors reserve the right to request original versions of figures and the original images that were used to assemble the figure.

We realize that it is difficult to revise to a specific deadline. In the interest of protecting the conceptual advance provided by the work, we recommend a revision within 3 months (19th Feb 2026). Please discuss the revision progress ahead of this time with the editor if you require more time to complete the revisions. Use the link below to submit your revision:

Referee #1:

The authors have addressed most of my questions. However, I have minor suggestions regarding the following two aspects:

1. The authors analyzed the conservation of the ZHD8 among 16 fruit crops, and the results indicated relatively sequence conservation of ZHD8 across different species. While these results are presented in the response letter, it is suggested that the authors retain this part of the findings in the main text. Additionally, the discussion section could discuss the prospect of ZHD8 potentially providing insights for the improvement of other crops.
2. A cartoon illustration of tomato is included in Fig 7, however, it fails to adequately reflect the key characteristics of the tomato plant. Specifically, the fruits are depicted as growing at the petiole. To avoid ambiguity, it is recommended that the authors replace this cartoon illustration to better display the growth characteristics of the tomato plant.

Referee #2:

I have reviewed the revised manuscript and remain concerned about the issues I raised earlier, which I believe have not been sufficiently addressed. The authors assert that 0.1% NaCl induces metabolic changes without impacting plant growth. However, the fruit weight of wild-type plants appears to be influenced by the salt treatment.

Moreover, as described in the revised methods, tomato plants were irrigated weekly with 0.1% NaCl solution from the flowering stage until fruit ripening—a duration sufficient for significant salt accumulation in the soil. To prevent any potential misunderstanding among readers, it is essential for the authors to determine the final NaCl concentration in the soil at the fruit ripening stage and evaluate the plants' salt tolerance during this period. This evaluation should include, for instance, phenotypic characterization, measurements of root and shoot biomass, quantification of Na⁺ and K⁺ contents in both roots and shoots, and analysis of soluble sugar content.

The decrease in fruit weight observed in wild-type plants under salt treatment (Figures 2G, 3M) lacks statistical significance, unlike the clear effects shown in other figures. To substantiate these findings, it is paramount that fruit yield under salt stress be evaluated with a sufficient number of independent replicates to ensure reliability.

Furthermore, while Figure 5G indicates varying salinity tolerance between *sizhd8* and *sweet12* under 0.6% Na⁺ treatment, the corresponding fruit phenotypes remain unexamined. Assessing these phenotypes is crucial for a comprehensive understanding of the observed tolerance differences.

Response to Editor and Reviewers

Dear Editor and Reviewers,

We are delighted by your positive feedback and appreciate your encouraging assessment of our work. We have thoroughly considered all your suggestions, which we find highly valuable for improving the manuscript. In this revised version, we have addressed each comment point by point and incorporated corresponding improvements. We are sincerely grateful for your insightful guidance, which has been instrumental in refining this work.

To Editor,

We have now received re-review reports from two referees, which I have included below. As you will see, you have addressed many of their concerns satisfactorily; however, I would like you, in response to Referee #2, to (in the main text) acknowledge and discuss the limitations of your work in the context of sample size and the potential for accumulation of NaCl in the soil (in the absence of these measurements being carried out). Please also consider the minor revisions recommended by Reviewer #1. Before I can finally accept the manuscript, there are also some remaining editorial points which need to be addressed. In this regard would you please:

Thank you for your guidance. We have completed these tasks in the revised draft and the response comments.

- remove figures from the main manuscript and upload them as individual, high-resolution figure files; ensure track changes is not used,

We have performed the related operations as you requested.

- state employment by and/or association with a biotech company in the

disclosure and competing interests statement,

In preparing our disclosure statement, we consulted the "Disclosure and competing interests statement" sections of numerous articles published in The EMBO Journal (Refer to the following picture). Based on this, we confirmed that "The authors declare no competing interests." We can affirm that there are no conflicts of interest associated with this work. Should you have specific concerns that require a different form of disclosure, please provide us with the relevant template or guidance for our reference.

The EMBO Journal

Gianluca Figlia et al

Wolfson RL, Chantranupong L, Wyant GA, Gu X, Orozco JM, Shen K, Condon KJ, Petri S, Kedir J, Scaria SM et al (2017) KICSTOR recruits GATOR1 to the lysosome and is necessary for nutrients to regulate mTORC1. *Nature* 543:438-442

Wolfson RL, Sabatini DM (2017) The dawn of the age of amino acid sensors for the mTORC1 pathway. *Cell Metab* 26:301-309

Woodard LE, Wilson MH (2015) piggyBac-ing models and new therapeutic strategies. *Trends Biotechnol* 33:525-533

Wu CC, Peterson A, Zinshteyn B, Regot S, Green R (2020) Ribosome collisions trigger general stress responses to regulate cell fate. *Cell* 182:404-416.e414

Yang H, Jiang X, Li B, Yang HJ, Miller M, Yang A, Dhar A, Pavletich NP (2017) Mechanisms of mTORC1 activation by RHEB and inhibition by PRAS40. *Nature* 552:368-373

Ye J, Kumanova M, Hart LS, Sloane K, Zhang H, De Panis DN, Bobrovnikova-Marjon E, Diehl JA, Ron D, Koumenis C (2010) The GCN2-ATF4 pathway is critical for tumour cell survival and proliferation in response to nutrient deprivation. *EMBO J* 29:2082-2096

Ye J, Palm W, Peng M, King B, Lindsten T, Li MO, Koumenis C, Thompson CB (2015) GCN2 sustains mTORC1 suppression upon amino acid deprivation by inducing Sestrin2. *Genes Dev* 29:2331-2336

Yuan W, Guo S, Gao J, Zhong M, Yan G, Wu W, Chao Y, Jiang Y (2017) General control nonderepressible 2 (GCN2) kinase inhibits target of rapamycin complex 1 in response to amino acid starvation in *Saccharomyces cerevisiae*. *J*

Project administration; Writing—review and editing. Sandra Müller: Investigation. Fabiola Garcia-Cortizo: Conceptualization; Investigation. Marilena Neff: Investigation. Glynis Klink: Data curation; Formal analysis; Investigation; Methodology. Gernot Poschet: Conceptualization; Formal analysis; Supervision; Investigation; Methodology; Project administration. Aurelio A Telean: Conceptualization; Supervision; Funding acquisition; Visualization; Writing—original draft; Project administration; Writing—review and editing.

Source data underlying figure panels in this paper may have individual authorship assigned. Where available, figure panel/source data authorship is listed in the following database record: [biostudies:S-SCDT-10_1038-544318-025-00505-1](https://biostudies.org/studies/S-SCDT-10_1038-544318-025-00505-1)

Funding

Open Access funding enabled and organized by Projekt DEAL.

Disclosure and competing interests statement

The authors declare no competing interests.

Open Access This article is licensed under a Creative Commons Attribution 4.0 International License, which permits use, sharing, adaptation, distribution and reproduction in any medium or format, as long as you give appropriate credit to the original author(s) and the source, provide a link to the Creative Commons licence, and indicate if changes were made. The images or other third party

Downloaded from <https://www.embopress.org>

- acknowledge funding by the National Natural Science Foundation of China (32573008) and Taishan Scholars Program (tsqn202306139) both in our online submission system and the manuscript's "Acknowledgements" section,

We have performed the related operations as you requested.

- reduce the number of keywords to five,

We have performed the related operations as you requested.

- use the first ten names + et al. in the reference section for longer author lists,

We have revised the reference format of the entire text in accordance with the style requirements of The EMBO Journal.

- remove the AC/CrediT section from the manuscript text,

We have removed the section of Author Contribution.

- include a callout for Figure S14 (as Appendix Figure S14) in the main text,

We have added the callout.

- complete the middle column (D) in our author checklist,

We have revised the author checklist. Please be aware that this list may not be exhaustive. If you identify any omissions, we would appreciate it if you could take a screenshot to indicate the specific location and notify us by email. We apologize for any inconvenience this may have caused.

- update source file names, titles, legends and manuscript callouts to Dataset EV1-EV10 instead of Dataset S1-S10; they should be uploaded individually as Dataset files with legends in a separate tab/sheet in each Excel file,

We have performed the related operations as you requested.

- convert the Appendix file to a PDF format; Supplementary Figures should be compiled in Appendix PDF - the title page should contain "Appendix for 'How salt stress sweetens tomatoes: root-derived ABA triggers a novel sugar accumulation cascade'" and a table of contents included with the page numbers for the listed items; nomenclature should be Appendix Figure Sx and Appendix Table Sx throughout the manuscript and Appendix PDF,

We have placed the previous Supplementary Figures in the "Expanded View Figures" section. Therefore, this study no longer includes the Appendix file.

- include a Reagents and Tools table,

We have added it in Methods.

- ensure dataset PRJNA1306750 is publicly available,

Yes, I am sure.

- provide a URL for PRJNA1306750 in the data availability statement,

We have added it in Methods.

- provide exact p values are not provided in the legends of figures 1B, C, D, E, F, G, I, J, K, L; 2A, B, C, D, E, F, G, H, I, J; 3 H-L, M-P; 4F, G, K, L, M; 5A-C,

D-F, H-K; 6D, G, H, I, J, M, N,

Under the interaction model of stress effect and gene effect, since one-way ANOVA can only display results with $P < 0.05$, we introduced exact P -values to demonstrate the magnitude of different genotypes' responses to salt stress. However, for other data where such detailed information is unnecessary, we adopted the concise asterisk notation to indicate statistical significance. This approach maintains figure clarity while effectively conveying core information, aligning with the requirements of most journals, including EMBO J (as verified through our examination of several published articles). Nevertheless, to address potential inquiries from particular readers, all exact P -values for our data are accessible in the Source Data. Therefore, we have retained the current presentation style to communicate the core findings of our research in the most concise and clear manner.

If you insist that the P value be included in the figure legends, please send me a sample of an article similar to this one. We will make the revisions according to your requirements and following the format of the sample.

- specify the statistical test that was used for data analysis in the legend of figure 4D,

We have added it in legend of Figure 4D and others.

- define box plots in terms of minima, maxima, centre, bounds of box and whiskers, and percentile in the legends of figures 1B, C, I, J, K, L; 2C, D, E, G, H, I, J; 3M-P; 5A-F; 6G-J,

We have added them in figure legends

- correct the section order as follows: Title page - Abstract - Keywords - Introduction - Results - Discussion - Methods - Data Availability - Acknowledgements - Disclosure and Competing Interests Statement - References - Figure Legends - Table(s) - Expanded View Figure Legends.

We have performed the related operations as you requested.

We include a synopsis of the paper (see <http://emboj.embopress.org/>). Please

provide me with a general summary image, a two sentence statement and 3-5 bullet points that capture the key findings of the paper.

We have added it.

To Referee #1,

The authors have addressed most of my questions. However, I have minor suggestions regarding the following two aspects:

1. The authors analyzed the conservation of the ZHD8 among 16 fruit crops, and the results indicated relatively sequence conservation of ZHD8 across different species. While these results are presented in the response letter, it is suggested that the authors retain this part of the findings in the main text. Additionally, the discussion section could discuss the prospect of ZHD8 potentially providing insights for the improvement of other crops.

Thank you for your suggestion. We have retained the conservative analysis of ZHD8 among 16 fruit crops and the related discussions (Fig. EV16A).

2. A cartoon illustration of tomato is included in Fig 7, however, it fails to adequately reflect the key characteristics of the tomato plant. Specifically, the fruits are depicted as growing at the petiole. To avoid ambiguity, it is recommended that the authors replace this cartoon illustration to better display the growth characteristics of the tomato plant.

Thank you for your suggestions. In the original draft, the growth characteristics of the tomato plants were inaccurately depicted in the cartoon illustration. Based on your feedback, we have developed two sets of model images, as shown below: the cartoon tomato (left) and the realistic tomato (right). By comparing the visual effects of the two models, we find the cartoon version on the left more appealing, particularly due to its fresh layout and clean, simple artistic approach. Therefore, we have selected the left image for the revised version. We trust you will find this choice agreeable.

To Referee #2,

1. I have reviewed the revised manuscript and remain concerned about the issues I raised earlier, which I believe have not been sufficiently addressed. The authors assert that 0.1% NaCl induces metabolic changes without impacting plant growth. However, the fruit weight of wild-type plants appears to be influenced by the salt treatment.

Thank you for your valuable comments regarding the NaCl concentration used in this study. We would like to clarify the following points to better justify the rationale behind our experimental design:

(1) Soils with a salt content of 0.1%–0.3% are classified as mildly saline. Under such conditions, most glycophytes, including tomato, typically exhibit some degree of growth inhibition and yield reduction.

(2) The objective of this study was to address scientifically meaningful questions under salt stress conditions relevant to actual tomato production. Based on industry experience and preliminary experiments, we selected 0.1% NaCl as an appropriate stress level.

(3) As shown in Fig. EV1, the 0.1% NaCl treatment resulted in a statistically significant ($P < 0.05$) reduction in growth and yield. However, from a practical agricultural perspective, this level of reduction is considered acceptable, as it is accompanied by improved fruit flavor and higher market value—traits that are economically advantageous for producers.

Therefore, the observed decrease in yield under salt stress in our study

aligns with both agronomic practice and physiological expectation. Meanwhile, the editor also requested that we, based on your suggestions, provide discussions and statements regarding the shortcomings of this part of the research. The relevant content has been revised and placed at the end of the conclusion. We hope this clarification addresses your concern.

2. Moreover, as described in the revised methods, tomato plants were irrigated weekly with 0.1% NaCl solution from the flowering stage until fruit ripening—a duration sufficient for significant salt accumulation in the soil. To prevent any potential misunderstanding among readers, it is essential for the authors to determine the final NaCl concentration in the soil at the fruit ripening stage and evaluate the plants' salt tolerance during this period. This evaluation should include, for instance, phenotypic characterization, measurements of root and shoot biomass, quantification of Na⁺ and K⁺ contents in both roots and shoots, and analysis of soluble sugar content.

Thank you for your question. We direct your attention to the "Salt stress treatment" section in the Materials and Methods chapter, where we detailed the salt stress treatment protocol. To reiterate: "Upon reaching the flowering stage, tomato plants were irrigated weekly either with water (Control) or with 0.1% NaCl solution (Salt stress). Each irrigation was applied until the soil was fully saturated. The salt stress treatment was maintained throughout the entire growth cycle of the tomatoes."

Thank you for raising this point, which allows us to clarify the irrigation protocol. The tomatoes were cultivated in large, bottom-perforated containers. The phrase "Each irrigation was applied until the soil was fully saturated" indicates that a sufficient volume of NaCl solution was applied to allow drainage from the bottom of the pot. This method effectively maintained a stable soil salt content of approximately 0.1%, thereby preventing excessive salt accumulation. We acknowledge that our initial description may have been unclear and have therefore revised the 'Materials and Methods' section to detail the salt stress treatment more precisely, which we believe will better

convey the rationale of our experimental design.

Regarding the salt tolerance indicators, please see Fig. EV1 and EV15, including (A) Phenotype of plants under salt stress. (B) Plant height. (C) Shoot dry weight. (D) Root dry weight. (E) Fruit yield per plant. (F) Net photosynthetic rate of the leaves. (G) Chlorophyll content of the leaves. (H) Nitrogen content of leaves. (I) Phosphorus content of leaves. (J) Potassium content of leaves. (K) Sodium content of leaves. (L) Hydrogen peroxide staining by 3,3'-diaminobenzidine. (M) Superoxide anion staining by nitroblue tetrazolium.

3. The decrease in fruit weight observed in wild-type plants under salt treatment (Figures 2G, 3M) lacks statistical significance, unlike the clear effects shown in other figures. To substantiate these findings, it is paramount that fruit yield under salt stress be evaluated with a sufficient number of independent replicates to ensure reliability.

Your question might be why the fruit weight of WT shows a significant difference between the Salt and Control treatments in Fig. 1B, but not in Fig. 2G and 3M. Let me explain this using the data analysis process from Fig. 1B and Fig. 2G.

(1) Fig. 1B employs a t-test analysis approach (see Fig. Q&A-1), which emphasizes the relationship between the two groups without considering other factors. If we analyze the WT data in Fig. 2G using the same t-test method, the difference is also significant (see Fig. Q&A-2). In other words, the results of this study are consistent under the same analysis method.

配对的 t 检验	
1	表进行了分析：
2	
3	列 B
4	vs.
5	列 A
6	
7	配对的 t 检验
8	P value
9	P value summary
10	显著不同 (p <0.05)?
11	一尾还是双尾 p 值?
12	t, df
13	
14	差异有多大?
15	Mean of 列 A
16	Mean of 列 B
17	手段之间的差异 (B - A) ± SEM
18	95% confidence interval
19	r 平方 (eta 平方)
20	
21	F 测试比较差异
22	F, DFn, Dfd
23	P value
24	P value summary
25	显著不同 (p <0.05)?
26	
27	数据分析：

Fig. Q&A-1

配对的 t 检验	
1	表进行了分析：
2	
3	列 B
4	vs.
5	列 A
6	
7	配对的 t 检验
8	P value
9	P value summary
10	显著不同 (p <0.05)?
11	一尾还是双尾 p 值?
12	t, df
13	
14	差异有多大?
15	Mean of 列 A
16	Mean of 列 B
17	手段之间的差异 (B - A) ± SEM
18	95% confidence interval
19	r 平方 (eta 平方)
20	
21	F 测试比较差异
22	F, DFn, Dfd
23	P value
24	P value summary
25	显著不同 (p <0.05)?

Fig. Q&A-2

(2) This is why we initially used the two-way ANOVA analysis method in the original manuscript. Under this analysis method, we focused on and compared the differences between Salt and Control treatments within the same genotype background. As shown in the results of our original manuscript, the difference between Salt and Control treatments for WT was also significant (see Fig. Q&A-3).

Fig. Q&A-3

(3) However, during the first revision, the reviewers asked us to comprehensively consider both the stress effect and the gene effect. This led us to change the original two-way ANOVA analysis method to the one-way ANOVA analysis method you see now. Due to the introduction of the gene effect of *SISnRK2.6*, the distribution range of the entire dataset expanded (especially the extreme low values). In short, the gene effect overshadowed the stress effect. As a result, the differences are no longer as significant (see Fig. Q&A-4).

Fig. Q&A-4

(4) For the above reasons, we also deliberated on which analysis method would be best. From a personal perspective, I advocate analyzing whether the Salt treatment has a significant impact compared to the Control treatment within the same genotype (i.e., two-way ANOVA). However, the reviewers' perspective is also valid, and the method of analyzing the interaction between

genotype and environment is widely used in many research papers. Therefore, we ultimately adopted the reviewers' suggestion (i.e., one-way ANOVA).

(5) Finally, I would like to emphasize that in scientific research, statistical significance is not always necessary for a finding to be meaningful. Qualitative changes result from the accumulation of quantitative changes. Even changes that are not statistically significant deserve our attention and recognition. Recently, Nature also published articles discussing and evaluating the adverse impacts of overemphasizing statistical significance in scientific research (see Editorial, 2023, Points of significance, *Nat Hum Behav* 7: 293-294).

Therefore, we hope that through these explanations and the reconstruction of the analysis process, we can address your concerns. We also hope that you will support the conclusions of this study. Thank you once again.

4. Furthermore, while Figure 5G indicates varying salinity tolerance between *sizhd8* and *sweet12* under 0.6% Na⁺ treatment, the corresponding fruit phenotypes remain unexamined. Assessing these phenotypes is crucial for a comprehensive understanding of the observed tolerance differences.

High-concentration salt stress is a standard method for assessing plant salt tolerance. For example, Wang et al. (2020, *The EMBO Journal*, 39: e103256) employed a 200 mM NaCl solution in their GWAS of tomato salt tolerance. In line with this approach, we used a 0.6% NaCl treatment to effectively screen for salt tolerance based on our previous research (Gong et al., 2013, *Scientia Horticulturae* 157: 1-12). However, as this concentration prevented the plants from completing their life cycle (including flowering and fruiting), our evaluation was necessarily confined to the seedling stage. Salt (0.1% NaCl) tolerance assessment of the entire growth period are primarily presented in Figures EV1 and EV15.

In the end, we believe the insightful suggestions you have provided address questions that future readers may also have. Accordingly, in the

revised manuscript, we have thoroughly refined the Methods and Discussion sections, including a clearer articulation of the study's limitations. We hope these revisions now more accurately reflect the original aims and conclusions of our work.

Thank you once again for your support and constructive input.

Sincerely,

Biao Gong

Dear Biao,

I am pleased to inform you that your manuscript has been accepted for publication in the EMBO Journal.

Congratulations! I am really glad to see this work published in the EMBO Journal.

You may qualify for financial assistance for your publication charges - either via a Springer Nature fully open access agreement or an EMBO initiative. Check your eligibility: <https://link.springer.com/journal/44318/how-to-publish-with-us>

Yours sincerely,

William

William Teale, PhD
Editor
The EMBO Journal
w.teale@embojournal.org

Please note that it is The EMBO Journal policy for the transcript of the editorial process (containing referee reports and your response letters) to be published as an online supplement to each paper. If you should prefer removal of any referee-only figures included in the point-by-point response(s), e.g. because they may still be used for future publication or because they have been reproduced from published work by others, please do let us know immediately via response email.

More information is available here: <https://link.springer.com/partners/embo-press/editorial-policies#Peer%20review>